# Reconstruction of transmission chains of SARS-CoV-2 amidst multiple outbreaks in a geriatric acute-care hospital: a combined retrospective epidemiological and genomic study

**Mohamed Abbas[1,2,3]\*, Anne Cori[2,4], Samuel Cordey[3,5], Florian Laubscher[5], Tomás Robalo Nunes[1,6], Ashleigh Myall[7,8], Julien Salamun[9], Philippe Huber[10], Dina Zekry[10], Virginie Prendki[10,11], Anne Iten[1], Laure Vieux[12], Valérie Sauvan[1], Christophe E Graf[10], Stephan Harbarth[1,3,11]**

[1]Infection Control Programme & WHO Collaborating Centre on Patient Safety, Geneva University Hospitals, Geneva, Switzerland; [2]MRC Centre for Global Infectious Disease Analysis, Imperial College London, London, United Kingdom; [3]Faculty of Medicine, University of Geneva, Geneva, Switzerland; [4]Abdul Latif Jameel Institute for Disease and Emergency Analytics (J-IDEA), School of Public Health, Imperial College London, London, United Kingdom; [5]Laboratory of Virology, Department of Diagnostics, Geneva University Hospitals, Geneva, Switzerland; [6]Serviço de Infecciologia, Hospital Garcia de Orta, EPE, Almada, Portugal; [7]Department of Infectious Diseases, Imperial College London, London, United Kingdom; [8]Department of Mathematics, Imperial College London, London, United Kingdom; [9]Department of Primary Care, Geneva University Hospitals, Geneva, Switzerland; [10]Department of Rehabilitation and Geriatrics, Geneva University Hospitals, Geneva, Switzerland; [11]Division of Infectious Diseases, Geneva University Hospitals, Geneva, Switzerland; [12]Occupational Health Service, Geneva University Hospitals, Geneva, Switzerland

**\*For correspondence:**
mohamed.abbas@hcuge.ch

## Abstract:

**Background:** There is ongoing uncertainty regarding transmission chains and the respective roles of healthcare workers (HCWs) and elderly patients in nosocomial outbreaks of severe acute respiratory syndrome coronavirus 2 (SARS-CoV-2) in geriatric settings.

**Methods:** We performed a retrospective cohort study including patients with nosocomial coronavirus disease 2019 (COVID-19) in four outbreak-affected wards, and all SARS-CoV-2 RT-PCR positive HCWs from a Swiss university-affiliated geriatric acute-care hospital that admitted both Covid-19 and non-Covid-19 patients during the first pandemic wave in Spring 2020. We combined epidemiological and genetic sequencing data using a Bayesian modelling framework, and reconstructed transmission dynamics of SARS-CoV-2 involving patients and HCWs, to determine who infected whom. We evaluated general transmission patterns according to case type (HCWs working in dedicated Covid-19 cohorting wards: HCW$_{covid}$; HCWs working in non-Covid-19 wards where outbreaks occurred: HCW$_{outbreak}$; patients with nosocomial Covid-19: patient$_{noso}$) by deriving the proportion of infections attributed to each case type across all posterior trees and comparing them to random expectations.

**Results:** During the study period (1 March to 7 May 2020), we included 180 SARS-CoV-2 positive cases: 127 HCWs (91 $HCW_{covid}$, 36 $HCW_{outbreak}$) and 53 patients. The attack rates ranged from 10% to 19% for patients, and 21% for HCWs. We estimated that 16 importation events occurred with high confidence (4 patients, 12 HCWs) that jointly led to up to 41 secondary cases; in six additional cases (5 HCWs, 1 patient), importation was possible with a posterior probability between 10% and 50%. Most patient-to-patient transmission events involved patients having shared a ward (95.2%, 95% credible interval [CrI] 84.2%–100%), in contrast to those having shared a room (19.7%, 95% CrI 6.7%–33.3%). Transmission events tended to cluster by case type: $patient_{noso}$ were almost twice as likely to be infected by other $patient_{noso}$ than expected (observed:expected ratio 2.16, 95% CrI 1.17–4.20, p=0.006); similarly, $HCW_{outbreak}$ were more than twice as likely to be infected by other $HCW_{outbreak}$ than expected (2.72, 95% CrI 0.87–9.00, p=0.06). The proportion of infectors being $HCW_{covid}$ was as expected as random. We found a trend towards a greater proportion of high transmitters (≥2 secondary cases) among $HCW_{outbreak}$ than $patient_{noso}$ in the late phases (28.6% vs. 11.8%) of the outbreak, although this was not statistically significant.

**Conclusions:** Most importation events were linked to HCW. Unexpectedly, transmission between $HCW_{covid}$ was more limited than transmission between patients and $HCW_{outbreak}$. This finding highlights gaps in infection control and suggests the possible areas of improvements to limit the extent of nosocomial transmission.

**Funding:** This study was supported by a grant from the Swiss National Science Foundation under the NRP78 funding scheme (Grant no. 4078P0_198363).

## Editor's evaluation

Congratulations on this useful and technically impressive paper demonstrating that phylogenetic and epidemiologic data can be used in a retrospective cohort to reconstruct that chain of events in terms of case importation into a high risk geriatrics ward. The conclusion that HCW (healthcare worker) transmission in non-COVID wards was particularly important is critical for hospital epidemiologists. The methodology advances will hopefully push the field forward in terms of tracking outbreaks in various settings.

## Introduction

Nosocomial acquisition of severe acute respiratory syndrome coronavirus 2 (SARS-CoV-2) in geriatric institutions and long-term care facilities (LTCFs) may account for large proportions of all declared coronavirus disease 2019 (Covid-19) cases in many countries, and contribute substantially to morbidity and mortality (*Thiabaud et al., 2021*; *Bhattacharya et al., 2021*; *The Guardian, 2021b*; *The Guardian, 2021a*). Because the reservoir of SARS-CoV-2 in healthcare environments may contribute to amplifying the pandemic (*Knight et al., 2022*), we need to better understand transmission dynamics in these settings.

The terms healthcare-associated, hospital-onset, and nosocomial Covid-19 reflect the uncertainty around defining and distinguishing community- versus healthcare-acquired Covid-19 cases (*Abbas et al., 2021c*). Nevertheless, in some settings, such as LTCFs and nursing homes, these definitions are relatively straightforward. In other settings, such as those with a high patient turnover, or where patients are admitted from the community and both Covid-19 and non-Covid-19 cases are hospitalised in the same institution, defining, and more importantly detecting cases is crucial to avoid cross-contamination. Determining sources and transmission pathways of infection may thus help improve infection prevention and control (IPC) strategies.

The role of healthcare workers (HCWs) in nosocomial Covid-19 transmission dynamics is complex, as they can be victims and/or vectors of SARS-CoV-2 infection, and can acquire from or transmit to their peers and patients and the community (*Abbas et al., 2021b*; *Ellingford et al., 2021*). There is ongoing controversy and uncertainty surrounding the role of HCWs in infecting patients during nosocomial outbreaks, and findings from acute-care hospitals cannot be applied directly to LTCFs and geriatric hospitals (*Abbas et al., 2021a*; *Klompas et al., 2021*; *Lucey et al., 2021*; *Aggarwal et al., 2022*).

The aim of this study was to reconstruct transmission dynamics in several nosocomial outbreaks of SARS-CoV-2 involving patients and HCWs in a Swiss university-affiliated geriatric hospital that admitted both Covid-19 and non-Covid-19 patients during the first pandemic wave in Spring 2020.

## Methods

We performed a retrospective cohort study of all patients with nosocomial Covid-19 in four outbreak-affected wards, and all SARS-CoV-2 RT-PCR positive HCWs from 1 March to 7 May 2020. This study is reported according to the STROBE (*von Elm et al., 2007*) (Strengthening the Reporting of Observational Studies in Epidemiology) and ORION (*Stone et al., 2007*) (Outbreak Reports and Intervention studies Of Nosocomial infection) statements.

### Setting

The Hospital of Geriatrics, part of the Geneva University Hospitals (HUG) consortium, has 196 acute-care and 100 rehabilitation beds. During the first pandemic wave, a maximum of 176 acute-care beds were dedicated to admitting geriatric patients with Covid-19 who were not eligible for escalation of therapy (e.g., intensive care unit admission) (*Mendes et al., 2020*). During the same period, patients were also admitted for non-Covid-19 hospitalisations, and the rehabilitation beds were also open to patients convalescing from Covid-19. Beginning on 1 April 2020, we use RT-PCR to screen all patients on admission for SARS-CoV-2. Between 7 April 2020 and 30 May 2020, we screened patients in non-Covid wards. We encouraged HCWs from outbreak wards to undergo PCR testing on nasopharyngeal swabs between 9 and 16 April 2020, even if they were asymptomatic. We described additional IPC measures in Appendix 1.

### Definitions

We defined healthcare-associated (HA) Covid-19 by an onset of symptoms ≥5 days after admission in conjunction with a strong suspicion of healthcare transmission, in accordance with Swissnoso guidelines (*Swissnoso, 2021*). We classified patients with HA-Covid-19 as 'patient$_{noso}$', the others were assumed to be community-acquired or 'patient$_{community}$'. An outbreak was declared when ≥3 cases of HA-Covid-19 (HCWs and patients) with a possible temporal-spatial link were identified (*Swissnoso, 2021*). HCWs were included in the outbreak investigation if they had a positive RT-PCR for SARS-CoV-2. We classified HCWs as 'HCW$_{covid}$' if they worked in a Covid-19 cohorting ward admitting community-acquired cases, or 'HCW$_{outbreak}$' if they worked in a 'non-Covid' ward (i.e., not admitting community-acquired Covid-19 cases) in which nosocomial outbreaks occurred. HCWs worked in either one or another type of ward, except for one HCW who worked in both Covid and non-Covid wards (although the proportion and/or days worked in each type of ward is unknown). Six HCWs worked across multiple wards (e.g., on-call) and were attributed to Covid-wards for the analyses.

### Data sources

The data used for this study were from the sources described previously (*Abbas et al., 2021a*). First, we used prospectively collected data from the Swiss Federal Office of Public Health-mandated surveillance of hospitalised Covid-19 patients (*Thiabaud et al., 2021*). We also used prospectively collected data from HUG's Department of Occupational Health for symptom-onset data and the Department of Human Resources (HR) for HCW shifts. Dates of symptom onset were available for both patients and HCWs from each source, respectively.

### Descriptive epidemiology

We produced an epidemic curve using dates of symptom onset; where these were unavailable (e.g., asymptomatic cases), we imputed them with the median difference between date of symptom onset and date of nasopharyngeal swab.

### Microbiological methods

All Covid-19 cases in the outbreak were confirmed by RT-PCR on nasopharyngeal swabs. We performed SARS-CoV-2 whole-genome sequencing (WGS) using an amplicon-based sequencing

method to produce RNA sequences, as previously described (*Abbas et al., 2021a*) and summarised in Appendix 1.

## Phylogenetic analysis

We performed sequence alignment with MUSCLE (v3.8.31). We employed MEGA X (*Kumar et al., 2018*) using the Maximum Likelihood method and Tamura three-parameter model (*Tamura, 1992*) to conduct the evolutionary analyses. We integrated to the phylogenetic analysis all the complete genomes SARS-CoV-2 sequenced by the Laboratory of Virology (HUG) for the purposes of epidemiological surveillance in the community.

## Statistical analysis

We performed descriptive statistics with medians and interquartile ranges (IQRs), and counts and proportions, as appropriate.

## Reconstruction of transmission trees

We combined epidemiological and genomic data to reconstruct who infected whom using the R package `outbreaker2` (*Jombart et al., 2014*; *Campbell et al., 2018*), as described elsewhere (*Abbas et al., 2021a*) and in Appendix 1. Briefly, the model uses a Bayesian framework, combining information on the generation time (time between infections in an infector/infectee pair), and contact patterns, with a model of sequence evolution to probabilistically reconstruct the transmission tree (see Appendix 1).

Because formal contact tracing was limited during the study period, we constructed contact networks based on ward or room presence for patients based on their ward movements, and on HCW shifts obtained from HR. We defined a contact as simultaneous presence on the same ward on a given day (see Appendix 1). The manner by which `outbreaker2` handles these contacts is conservative in that it allows for non-infectious contacts to occur (false positives) and incomplete reporting of infectious contacts (false negatives). In addition, the model estimates the proportions of these contacts.

Using the reconstructed transmission trees, we determined the number of imported cases (with minimal posterior support of 10%), and the number of secondary cases generated by the imports. Imported caseswere defined as those that do not have apparent ancestors among the cases included in the outbreak. We also calculated the number of secondary (i.e., onward) infections for each case, that is, the individual reproductive number ($R$), which we stratified by epidemic phase (early or late with a cut-off on 9 April 2020) and case type ($HCW_{covid}$, $HCW_{outbreak}$, $patient_{noso}$, and $patient_{community}$, see Appendix 1).

We assessed the role of each case type in transmission by estimating the proportion of infections attributed to the case type ($f_{case}$), which we compared with the random expectation considering the prevalence of each case type (see Appendix 1). To better understand the transmission pathways between and within wards, we also estimated (for outbreak and non-outbreak wards), the proportion of infections attributed to infectors in the same ward. We also constructed a matrix representing ward-to-ward transmission. Patient movements between wards were constructed using the implementation of the vistime package (visualisation tool) as in the publication by *Meredith et al., 2020*. Statistical analyses were performed in R software version 4.0.3 (https://www.R-project.org/).

## Results

During the study period, we included a total of 180 SARS-CoV-2 positive cases: 127 HCWs of whom 91 $HCW_{covid}$, and 36 $HCW_{outbreak}$, and 53 patients from the four outbreak wards. Of the 53 included patients, post hoc epidemiological analysis showed that 4 of these likely acquired Covid-19 in the community (CA-Covid-19). The remaining 49 nosocomial cases represented 20.2% (49/242) of all patients hospitalised with Covid-19, and 81.7% (49/60) of nosocomial Covid-19 cases in the Geriatric Hospital. The ward-level attack rates ranged from 10% to 19% among patients. Moreover, 21% of all HCWs in the geriatric hospital had a PCR-positive test. The epidemic curve is shown in *Figure 1*, and ward-level epidemic curves in *Figure 1—figure supplement 1*. Characteristics of patients and HCWs are summarised in *Tables 1 and 2*, respectively. Strikingly, the time period between date of onset of

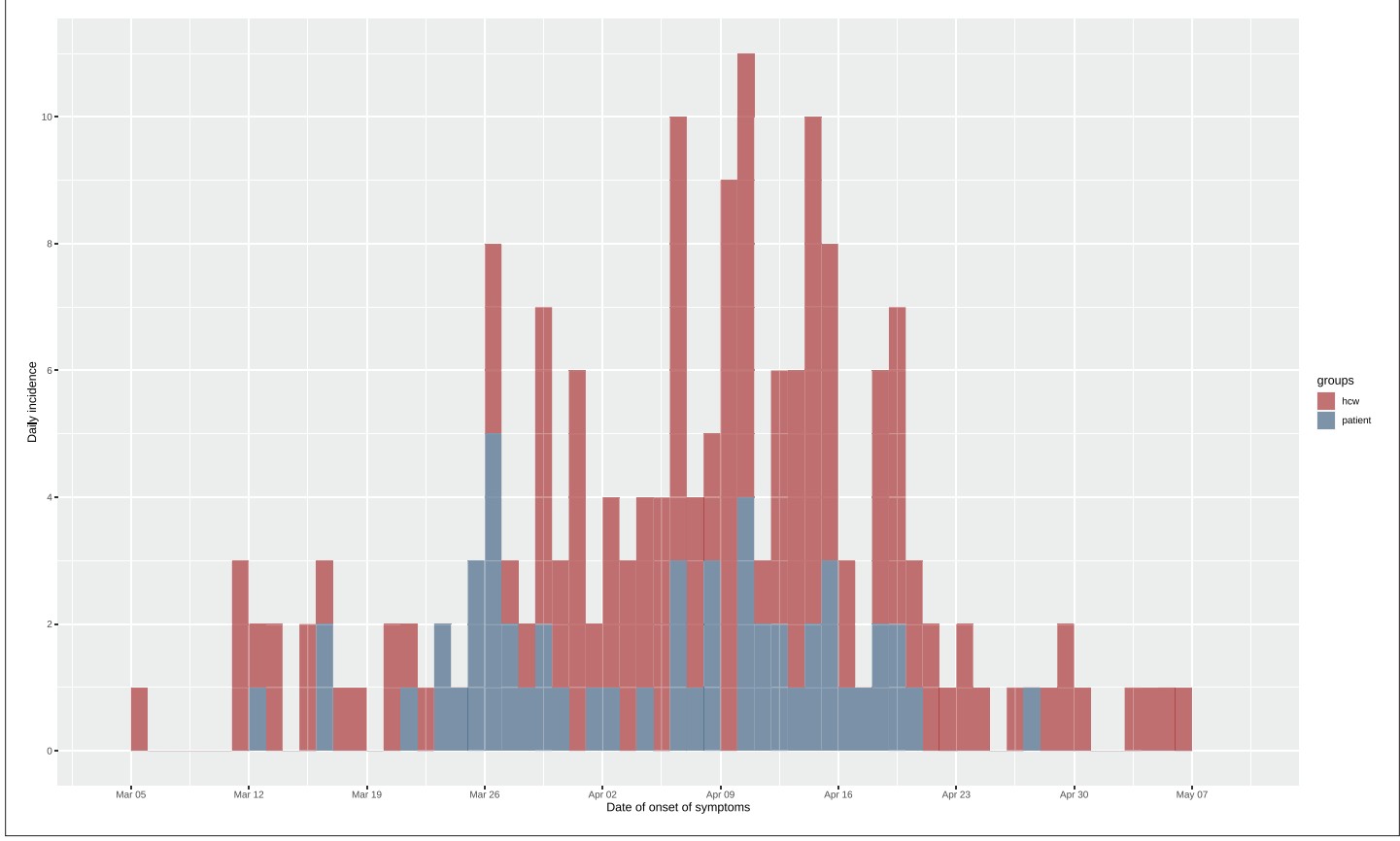

**Figure 1.** Epidemic curve of the nosocomial COVID-19 outbreak in a geriatric hospital involving HCWs and patients. Includes eight asymptomatic cases for whom date of onset was inferred (c.f., text). HCW, healthcare worker.

The online version of this article includes the following figure supplement(s) for figure 1:

**Figure supplement 1.** Ward-level epidemic curve.

symptoms and date of swab was shorter for $HCW_{covid}$ (mean 1.6 days, SD 1.78) than for $HCW_{outbreak}$ (mean 2.88 days, SD 4.84).

## Phylogenetic tree

We obtained SARS-CoV-2 sequences for 148 isolates of the 180 cases (82.2%), including 105 HCWs (82.7%) and 43 patients (81.1%). A rooted phylogenetic tree found substantial genetic diversity, with at least nine clusters and sub-clusters (*Figure 2*). One cluster (with moderate bootstrap support [BS] 26%) comprised sequences from 17 HCWs, 3 patients, and 6 community isolates in multiple

**Table 1.** Characteristics of Covid-19 patients with nosocomial acquisition.

| Characteristics | All patients (N=49) |
|---|---|
| Female, n (%) | 28 (57.1) |
| Age, median (IQR) | 85.4 (83.5–89.3) |
| | |
| Asymptomatic, n (%) | 3 (6.1) |
| Onset of symptoms before swab date, n (%) | 12 (24.5) |
| Days from onset of symptoms to swab, median (IQR) | 0 (0–0) |
| Days from onset of symptoms to swab, mean (SD) | –0.29 (2.19) |

**Table 2.** Characteristics of SARS-CoV-2 RT-PCR positive healthcare workers.

| Characteristics | All HCWs (N=127) |
|---|---|
| Female, n (%) | 92 (72.4) |
| Age, median (IQR) | 32.0 (43.3–54.8) |
| | |
| Profession, n (%) | |
| Nurse | 57 (44.9) |
| Nurse assistant | 39 (30.7) |
| Doctor | 19 (15.0) |
| Care assistant | 4 (3.2) |
| Transporter | 4 (3.2) |
| Physical therapist | 2 (1.6) |
| Speech therapist | 1 (0.8) |
| Medical student | 1 (0.8) |
| | |
| Asymptomatic, n (%) *missing data for 5* | 5 (3.9) |
| Days from onset of symptoms to swab, median (IQR) | 1 (−2 to 21) |
| HCWs in Covid-19 wards ($HCW_{covid}$) | 1 (1–2) |
| HCWs in non-Covid (outbreak) wards ($HCW_{outbreak}$) | 1 (0–3) |
| Days from onset of symptoms to swab, mean (SD) | 1.91 (2.86) |
| HCWs in Covid-19 wards ($HCW_{covid}$) | 1.60 (1.78) |
| HCWs in non-Covid (outbreak) wards ($HCW_{outbreak}$) | 2.88 (4.84) |

subclusters (e.g., BS 72% with signature mutation C5239T). Another cluster (BS 78% with signature mutations C28854T and A20268G) showed a HCW sequence (H1048) with high similarity with community isolates. There was also a large cluster (BS 68% with signature mutations C8293T, T18488C, and T24739C) with several subclusters includes 19 HCWs, 9 patients, and 3 community cases; ward movements for the patients are shown in *Appendix 1—figure 1*. A well-defined cluster (BS 100%) included isolates from patients and HCWs from the same ward.

## Imported cases

From the reconstructed trees, we identified 22 imports in total (17 HCWs, 5 patients) with posterior support ≥10%. The 22 imported cases generated 41 secondary cases (posterior support ≥10%), with a median posterior support of 32.4% (IQR 17.0%–53.7%). When restricting to imports with ≥50% posterior support, there were 16 imported cases 16 (12 HCWs, 4 patients), generating 35 secondary cases. There was some degree of uncertainty, reflected by circular transmission pathways, in determination of imported cases and their secondary cases. There were six transmission pairs (C114-C115, C153-H1057, H1008-H1059, H1011-H1019, H1017-H12021, and H1052-H1082) where there was uncertainty as to which of the cases was imported and which was a secondary case, that is, for each case in the pair there was a ≥10% probability of importation, but also ≥10% probability of being a secondary case of an imported case. Therefore, in total, 29 cases were 'pure' secondary cases of imported cases (*Table 3*).

## Reconstructing who infected whom

*Figure 3* shows the distribution of posterior support when considering the ancestry from the individual (i.e., Cxxx or Hxxx), case type ($HCW_{covid}$, $HCW_{outbreak}$, $patient_{noso}$, and $patient_{community}$), ward, or ward type (outbreak or non-outbreak wards). There was less confidence in attribution of ancestry

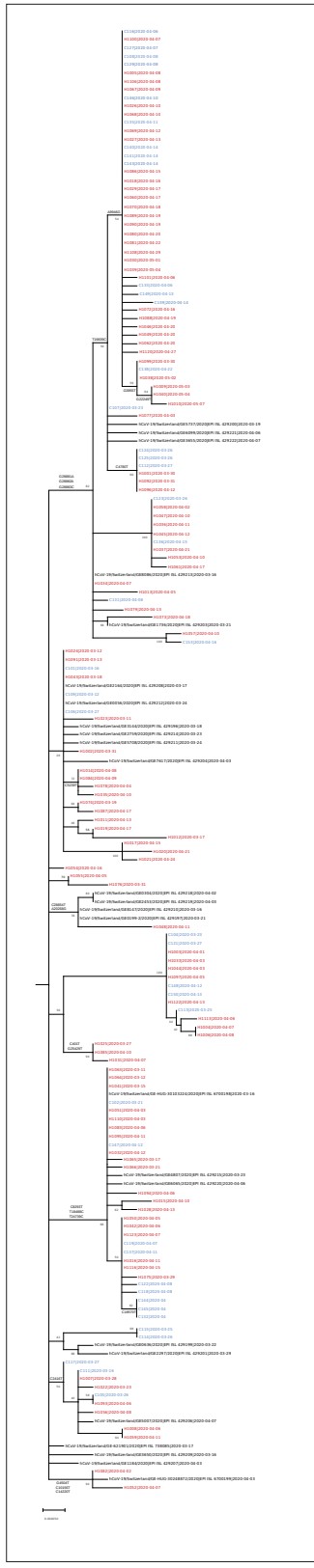

*Figure 2 continued*

alongside the community cases in the canton of Geneva, Switzerland, that were sequenced in March–April 2020 by the Laboratory of Virology (Geneva University Hospitals) and submitted to GISAID (virus names and accession ID [i.e., EPI_ISL_] are indicated) in the context of an epidemiological surveillance. For each sequence the date of the sample collection is mentioned (yyyy-mm-dd).

when considering by individual or ward, when compared with case type or ward type. The output from the ancestry reconstruction is shown in *Figure 3—figure supplement 1* and *Figure 3—figure supplement 1—source code 1*. The model estimated that the reporting probability was 91.2% (95% credible interval [CrI] 86.6%–95.2%), suggesting that only 8.8% of source cases involved in transmission were not identified. For most (90.8%) cases, the model identified the direct infector, without intermediate unobserved cases (*Figure 3—figure supplement 2*).

## Ward attribution

The epidemiological attribution of the presumptive ward on which patients became infected and that suggested by the model output (see Appendix 1) agree for 95% of the nosocomial cases. The modelling analysis modified the ward attribution for three patients.

## Transmission patterns

Among patient-to-patient transmission events, and across all posterior trees, 95.2% (95% CrI 84.2%–100%) involved patients who shared a ward during their hospital stay. In contrast, only 19.7% (95% CrI 6.7%–33.3%) of patient-to-patient transmissions involved patients who had shared a room. The model predicted that C107 infected C131 with a 72.5% probability although they did not share a ward (*Appendix 1—figure 1B* for ward movements); the probabilities that this was a direct infection and indirect infection with an unreported intermediate infector were 38.3% and 34.2%, respectively (*Appendix 1—figure 1B* for ward movements).

## Secondary infections

The number of secondary infections caused by each infected case (individual reproductive number *R*, estimated from the transmission tree reconstruction), ranged from 0 to 9 (*Figure 4*). We compared the proportion of cases with no secondary transmissions (non-transmitters) and of cases with ≥2 secondary transmissions (high

**Figure 2.** Phylogenetic tree of SARS-CoV-2 genome sequences. The tree includes 148 sequences related to the outbreak (patient and employee sequences are named C1xx [blue] and H10xx [red], respectively),

**Table 3.** Imported cases and secondary infections, patients and HCWs are named C1xx and H10xx, respectively.

| Imported case | Posterior probability of importation | Secondary onward transmission by imported case | Posterior probability of onward transmission |
|---|---|---|---|
| C107 | 100 | H1077<br>C131<br>C124<br>H1005<br>C125<br>H1034<br>H1068<br>C112<br>C116 | 100<br>72.5<br>39.4<br>35.0<br>32.9<br>27.3<br>18.5<br>15.9<br>11.7 |
| C114* | 42.5 | C115* | 42.5 |
| C115* | 57.5 | C114* | 57.5 |
| C123 | 96.4 | H1058<br>H1036<br>H1047 | 90.1<br>16.0<br>11.1 |
| C153* | 51.7 | H1057* | 51.6 |
| H1008* | 85.7 | H1059* | 85.7 |
| H1011* | 65.9 | H1019* | 61.7 |
| H1012 | 100 | N/A | N/A |
| H1013 | 100 | N/A | N/A |
| H1015 | 100 | N/A | N/A |
| H1017* | 52.3 | H1020<br>H1021* | 100.0<br>52.3 |
| H1019* | 34.1 | H1011* | 28.2 |
| H1021* | 47.7 | H1017* | 47.7 |
| H1025 | 86.6 | H1085<br>H1031 | 95.7<br>41.5 |
| H1048 | 100 | N/A | N/A |
| H1052* | 18.9 | H1082* | 18.9 |
| H1057* | 48.3 | C153* | 48.3 |
| H1059* | 14.3 | H1008* | 14.3 |
| H1073 | 100 | N/A | N/A |
| H1082* | 81.1 | H1052* | 81.1 |
| H1110 | 85.1 | H1063<br>H1064<br>H1041<br>H1024<br>H1091<br>H1065 | 58.0<br>32.4<br>30.0<br>22.5<br>22.1<br>17.2 |
| H1122 | 84.5 | C113<br>C104<br>H1033<br>H1003<br>H1004<br>H1044 | 53.7<br>26.0<br>17.0<br>15.5<br>15.0<br>10.6 |

N/A: not applicable.

*Uncertainty in transmission (i.e., case could either be an imported case or a secondary case).

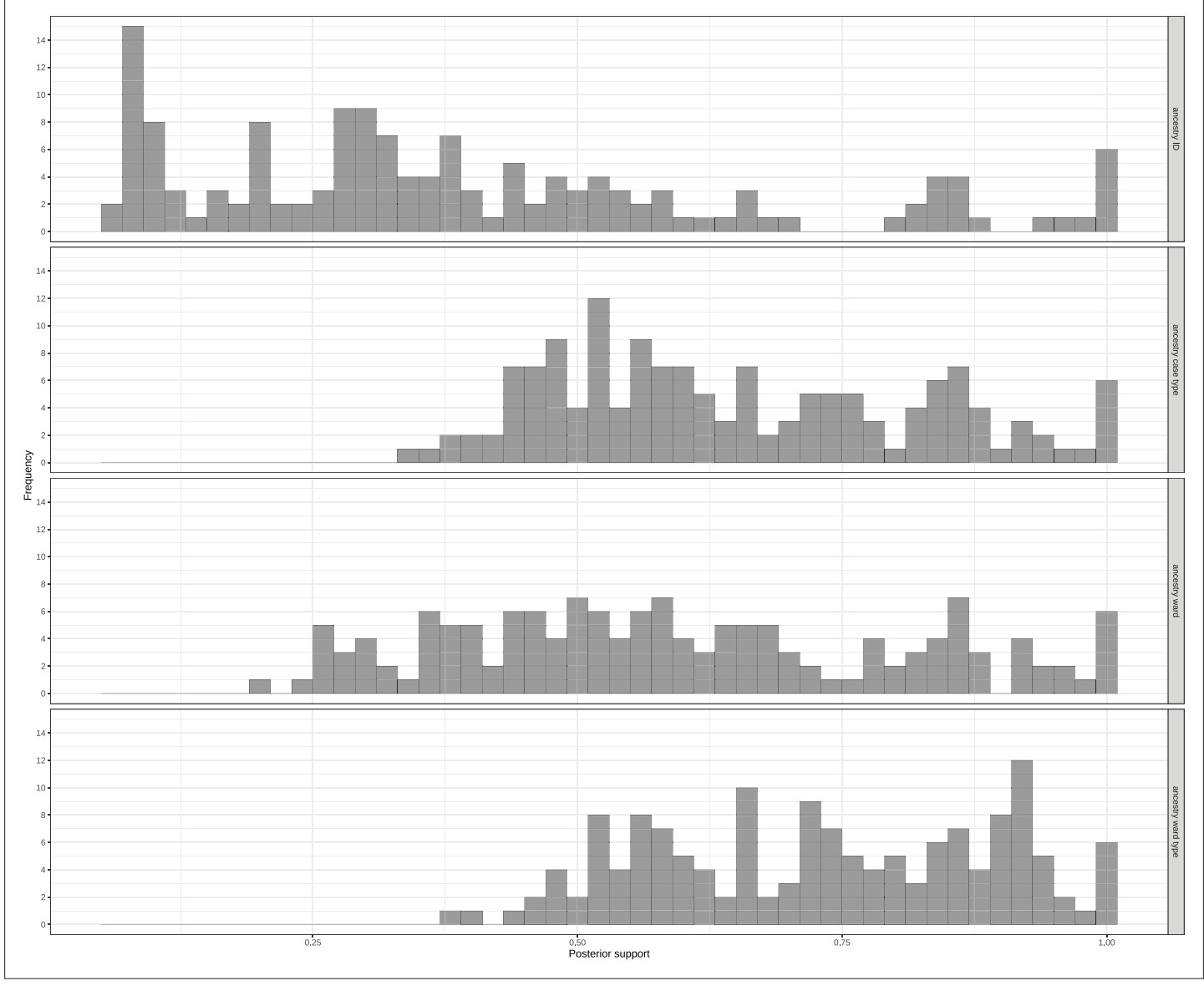

**Figure 3.** Distribution of posterior support of maximum posterior ancestry for all cases, according to identity of (**A**) individual ancestor, (**B**) ancestor's case type (i.e. , 'HCW$_{covid}$', 'HCW$_{outbreak}$', 'patient$_{noso}$', and 'patient$_{community}$'), (**C**) ancestor's ward, and (**D**) ancestor's ward type (i.e., 'outbreak ward', 'non-outbreak ward').

The online version of this article includes the following source code and figure supplement(s) for figure 3:

**Figure supplement 1.** Ancestry reconstruction (who infected whom) of the `outbreaker2` model.

**Figure supplement 1—source code 1.** Interactive ancestry plot (who infected whom) – identical to *Figure 3—figure supplement 1*.

**Figure supplement 2.** Distribution of number of missed generations across posterior trees, stratified by phase of outbreak.

**Figure supplement 3.** Comparison of the accuracy of ancestry attribution of each sensitivity analysis.

transmitters) across case types and outbreak phase. We found that the proportion of non-transmitters among both HCW$_{outbreak}$ and patient$_{noso}$ was smaller in the early than in the late stage (approximately 32% in early and 55% in late phase for both groups), suggesting that the contribution of these groups to ongoing transmission decreased over the study period. Conversely, the proportion of non-transmitters among HCW$_{covid}$ was stable at about 55%–60% across the early and the late phase. The proportions of high transmitters were higher among HCW$_{outbreak}$ than either patient$_{noso}$ or HCW$_{covid}$ in the late phases (28.6% vs. 11.8% and 13.9%) of the outbreak. However, due to small numbers,

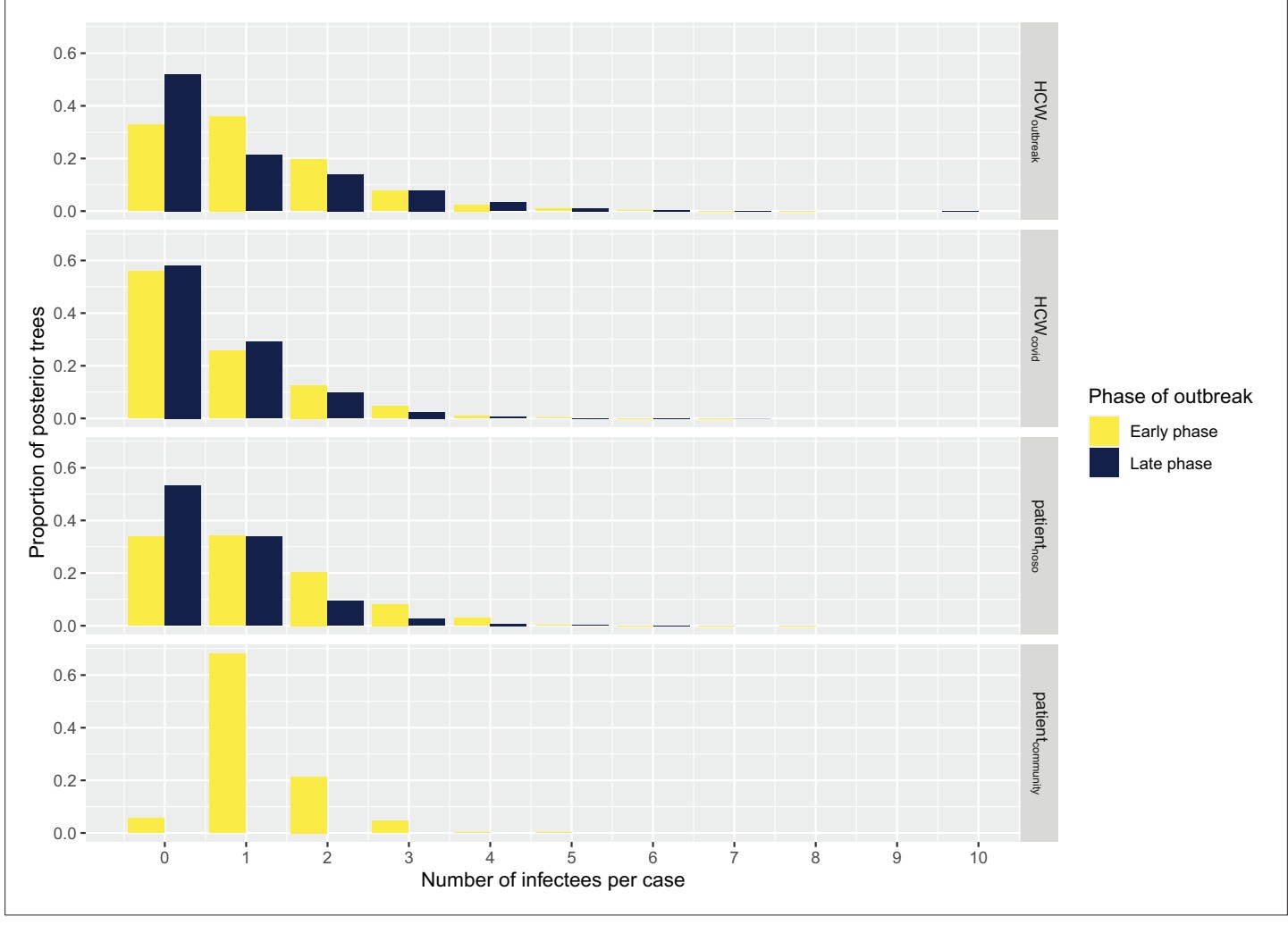

**Figure 4.** Histograms displaying the distributions of secondary cases by each case type ('HCW$_{covid}$', HCWs working in Covid-19 wards; 'HCW$_{outbreak}$', HCWs working in outbreak wards; 'patient$_{noso}$', patients with hospital-acquired Covid-19; 'patient$_{community}$', patients with community-acquired Covid-19) and stratified according to early (up to 9 April 2020) and late phases (as of 10 April 2020). Number of cases in early phase: HCW$_{outbreak}$ 19, HCW$_{covid}$ 43, patient$_{noso}$ 25, patient$_{community}$ 1. Number of cases in late phase: HCW$_{outbreak}$ 7, HCW$_{covid}$ 36, patient$_{noso}$ 17, patient$_{community}$ 0. HCW, healthcare worker.

these differences were not statistically significant. These trends were similar in the sensitivity analyses (*Appendix 1—table 2*).

## Role of HCWs and patients in transmission events

We found that cases were significantly less likely than expected at random to be infected by HCWs from COVID wards (proportion infected by HCW$_{covid}$, $f_{HCW}$=42%; 95% CrI 36%–49% vs. 53% expected at random; 95% CrI 44%–62%; p=0.042). This was true across all cases, but particularly among HCWs in outbreak wards ($f_{HCW}$=31%, 95% CrI 17%–48%; p=0.07) and patients ($f_{HCW}$=24%, 95% CrI 12%–37%, p=0.006). Conversely HCW$_{outbreak}$ were significantly more likely than expected at random to become infected by other HCW$_{outbreak}$ (proportion infected by HCW$_{outbreak}$, $f_{outbreak}$ = 38%, 95% CrI 22%–52% v. 18%; 95% CrI 4%–35%, p=0.03). Patients with nosocomial Covid-19 (patient$_{noso}$) were significantly more likely than expected at random to be infected by other patient$_{noso}$ (proportion infected by patient$_{noso}$, $f_{pat}$=56%, 95% CrI 41%–71% vs. 28%; 95% CrI 14%–45%, p=0.005). Full results are shown in *Figure 5*.

## Role of within-ward and between-ward transmission

Infected staff or patients in outbreak wards were responsible for significantly more transmission in general (54%; 95% CrI 48%–61%) than expected (43.7%, 95% CrI 35%–53%). This was driven in

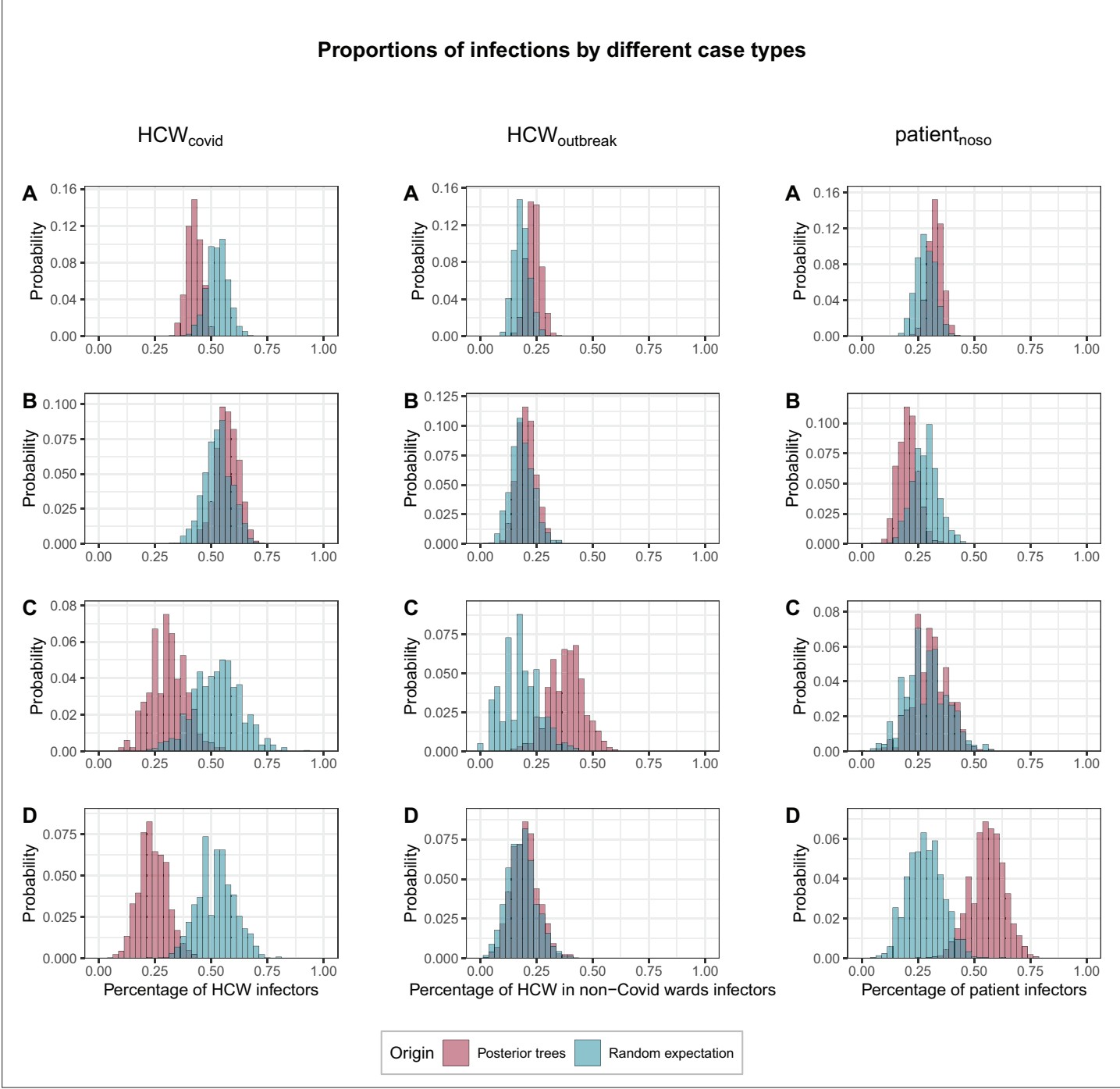

**Figure 5.** Proportions of transmissions ($f_{case}$) attributed to each case type (HCW$_{covid}$, HCW$_{outbreak}$, patient$_{noso}$, and patient$_{community}$) for each of the 1000 posterior trees retained. The blue histograms indicate the expected random distributions of $f_{case}$, given the prevalence of each case type. The red histograms show the observed distribution of $f_{case}$, across 1000 transmission trees reconstructed by `outbreaker2`. (**A**) All cases. (**B**) Transmission to HCWs in Covid-19 wards only. (**C**) Transmission to HCWs in non-Covid-19 wards (i.e., outbreak wards) only. (**D**) Transmission to patients with nosocomial Covid-19 only. HCW, healthcare worker.

particular by transmission from staff or patients to other staff or patients in outbreak wards (73%, 95% CrI 63%–82%) (***Figure 5***). Within-ward transmission was more pronounced in outbreak wards (mean 48%; range: 20%–70% of all infections within a ward) than in non-outbreak wards (mean 14%; range 0%–63%) as shown in ***Appendix 1—figures 2 and 3***.

## Discussion

This in-depth investigation of SARS-CoV-2 transmission between patients and HCWs in a geriatric hospital, including several nosocomial outbreaks, has provided many valuable insights on transmission dynamics. First, we showed that the combination of epidemiological and genetic data using sophisticated modelling enabled us to tease out overall transmission patterns. Second, we showed that transmission dynamics among HCWs differed according to whether they worked in Covid wards or in wards where outbreaks occurred. We found that HCW-to-HCW transmission in Covid wards was not higher than expected but the risk of transmission between HCWs in non-Covid wards was twofold higher than expected. Third, we identified excess patient-to-patient transmission events, most of which occurred within the same ward, but not necessarily the same room. Fourth, we identified multiple importation events that led to a substantial number of secondary cases or clusters; most were related to HCWs, but one was related to a patient with community-acquired Covid-19.

These results are particularly important, as settings which care for elderly patients, such as geriatrics and rehabilitation clinics or LTCFs have high attack rates of SARS-CoV-2 for both patients and HCWs (*Abbas et al., 2021b*). In an institution-wide seroprevalence study in our hospital consortium, the Department of Rehabilitation and Geriatrics, of which the hospital in this study was part, had the highest proportion of HCWs with anti-SARS-CoV-2 antibodies (*Martischang et al., 2022*).

The different transmission patterns between HCWs in Covid wards and outbreak wards (i.e., meant to be 'Covid-free') is intriguing. HCWs' behaviour may have affected transmission. Indeed, a previous study *Ottolenghi et al., 2021* found that HCWs caring for Covid-19 patients were concerned about becoming infected while caring for patients, and therefore may apply IPC measures more rigorously than when caring for patients who do not have Covid-19. HCWs working in non-Covid wards may not have felt threatened by Covid-19 patients who were in principle allocated to other wards, and thus not in their direct care. HCWs in non-Covid wards also may have underestimated the transmission risk from their peers to a greater extent than those working in Covid wards, and thus may not have maintained physical distancing well. In addition, we found a higher mean duration (2.9 days) of presenteeism despite symptoms compatible with Covid-19 among HCWs working in non-Covid wards than for those working in Covid wards (1.6 days), which gives credence to the abovementioned possible explanations for the different transmission patterns. Other factors (e.g., work culture, baseline IPC practices) also may have affected transmission patterns.

Most patient-to-patient transmission events involved patients who were hospitalised in the same ward, and were therefore in close proximity. We cannot exclude transmission from a 'point-source' or via a HCW's contaminated hands (i.e., an unidentified HCW who transmitted SARS-CoV-2 to multiple patients in the same ward). To date, we have little evidence to suggest that this is the case; indeed, the transmission patterns were robust to changes in the model assumptions. Mathematical models have suggested that single-room isolation of suspected cases could potentially reduce the incidence of nosocomial SARS-CoV-2 transmission by up to 35% (*Evans et al., 2021*). In the outbreaks we describe, symptomatic patients were identified promptly, with a median delay of 0 days between symptom onset and their first positive test. However, these precautions may not be sufficient as patients may transmit the virus when they are pre-symptomatic (*Ferretti et al., 2020*). Thus, infection prevention teams may need to identify patients at high risk of developing nosocomial Covid-19 (*Myall et al., 2021*) if single rooms are not available for all exposed patients (e.g., in cases of overcrowding). For example, a previous study *Mo et al., 2021* found that exposure to community-acquired cases who were identified and segregated or cohorted was associated with half the risk of infection compared with exposure to hospital-acquired cases or HCWs who may be asymptomatic. One possible explanation for this finding is that patients with CA-Covid may have passed the peak of infectiousness when they are admitted, whereas patients with HA-Covid cases have frequent unprotected contact with HCWs and other patients during their period of peak infectiousness.

The current evidence does not support the use of real-time genomics for control of SARS-CoV-2 nosocomial outbreaks (*Stirrup et al., 2021*; *Stirrup et al., 2022*). Nevertheless, in this investigation, as in many others, we performed WGS to investigate transmission patterns (*Aggarwal et al., 2022*). Although we were able to gain considerable insight from the powerful combination of genetic sequencing data and rich epidemiological data, we controlled the outbreaks without using WGS, as was the case in many published reports (*Arons et al., 2020*; *Taylor et al., 2020*). Furthermore, WGS may be more useful for ruling out transmission rather than for confirmatory purposes, due to the low

number of mutations accumulated in the SARS-CoV-2 genome between transmission pairs (**Braun et al., 2021**).

Our study has several strengths. To the best of our knowledge, it is the only extensive outbreak investigation that employed WGS and sophisticated modelling to assess SARS-CoV-2 transmission in a geriatric acute-care hospital. We performed WGS on isolates from a high proportion (80%) of cases, including those from HCWs which has been previously shown to improve understanding of transmission dynamics (**Ellingford et al., 2021**). In addition, we collected the data prospectively, thereby minimising the risk of bias.

Despite these strengths, some limitations must be acknowledged. First, we included only one sequence from a CA-Covid case. However the method we used to reconstruct who infected whom is able to cope with and identify missing intermediate cases, which allowed us to estimate that the overwhelming majority of cases (91.5%) was captured in our sample. Some misclassification may have occurred, for example, whether an imported cases is truly nosocomial or not. For example, the model predicted that patients C107 and C115 were imported cases. The predicted dates of infection were 14–22 March for case C107 and 8–24 March for case C115; their admission dates were 3 March and 18 March, respectively. The probability that infection occurred on or after date of admission for case C115 was 81%. So the fact that the cases are labelled as 'imported' does not preclude the fact that these were still nosocomial cases, simply that their infector was not identified in this outbreak. We did not collect data on adherence to Covid-specific IPC recommendations by HCWs in different wards. We were unable to relate the number of secondary infections with the population (ward) size; more complex models would be required to explain the underlying mechanisms in a context of fluctuating denominators, for example, due to ward closures, and so on. In addition, we performed these investigations during the first pandemic wave with a wild-type variant of SARS-CoV-2 in a susceptible population. For these reasons, the results may no longer be applicable in settings with high vaccination coverage and/or substantial natural immunity, or in later stages of the pandemic, also due to accrued experience in managing and preventing outbreaks. Nevertheless, the lessons learned may be useful in a large number of countries with slow vaccine roll-out due to vaccine hesitancy, particularly among HCWs where there is no vaccine mandate, or unequal access to vaccine supplies (**The Lancet Infectious Diseases, 2021**). Furthermore, nosocomial outbreaks of SARS-CoV-2 still occur despite high vaccination coverage (**Burugorri-Pierre et al., 2021**; **Shitrit et al., 2021**). Also, these valuable lessons may be applicable for nosocomial outbreak control in the case of future pandemics due to respiratory viruses with characteristics similar to SARS-CoV-2.

In conclusion, strategies to prevent nosocomial SARS-CoV-2 transmission in geriatric settings should take into account the potential for patient-to-patient transmission and the transmission dynamics between HCWs in non-Covid wards, which our study suggests may differ from those in dedicated Covid-19 wards.

## Acknowledgements

The authors would like to thank Frédéric Bouillot (Geneva University Hospitals) for providing us with HR data, Aurore Britan (Geneva University Hospitals) for data management, Rachel Goldstein for data collection (Geneva University Hospitals), Daniel Teixeira for data extraction (Geneva University Hospitals). In addition, the authors would like to thank Stéphanie Baggio, Frédérique Jacquerioz Bausch, Hervé Spechbach from the AMBUCoV study (Geneva University Hospitals), Amaury Thiabaud (University of Geneva), and the IPC team members from Geneva University Hospitals involved in SARS-CoV-2 outbreak management (Pascale Herault, Didier Pittet, Walter Zingg). Thibaut Jombart (London School of Hygiene and Tropical Medicine) and Finlay Campbell (World Health Organization) provided input on technical aspects of the `outbreaker2` model development. The study team also wishes to thank the dedicated HCWs who cared for the patients, and the Geriatric Hospital Covid-19 crisis management team (Gabriel Gold, Charline Couderc, Etienne Satin). This work was supported by a grant from the Swiss National Science Foundation under the NRP78 funding scheme (Grant no. 4078P0_198363). Mohamed Abbas and Anne Cori are supported by the National Institute for Health Research (NIHR) Health Protection Research Unit in Modelling and Health Economics, a partnership between Public Health England, Imperial College London and LSHTM (grant code NIHR200908). Anne Cori also acknowledges funding from the MRC Centre for Global Infectious Disease Analysis (reference MR/R015600/1), jointly funded by the UK Medical Research Council (MRC) and the UK Foreign,

Commonwealth & Development Office (FCDO), under the MRC/FCDO Concordat agreement and is also part of the EDCTP2 programme supported by the European Union. Ashleigh Myall was supported by a scholarship from the Medical Research Foundation National PhD Training Programme in Antimicrobial Resistance Research (MRF-145-0004-TPG-AVISO). The views expressed are those of the author(s) and not necessarily those of the NIHR, Public Health England or the Department of Health and Social Care.

## Additional information

### Competing interests

Anne Cori: received honoraria (which was paid to the institution) from Pfizer for lecturing on a course on mathematical modelling of infectious disease transmission and vaccination book. The author has no other competing interests to declare. The other authors declare that no competing interests exist.

### Funding

| Funder | Grant reference number | Author |
| --- | --- | --- |
| Schweizerischer Nationalfonds zur Förderung der Wissenschaftlichen Forschung | 4078P0_198363 | Mohamed Abbas Samuel Cordey Florian Laubscher Tomás Robalo Nunes Anne Iten Stephan Harbarth Anne Cori |
| National Institute for Health Research Health Protection Research Unit | NIHR200908 | Mohamed Abbas Anne Cori |

The funders had no role in study design, data collection and interpretation, or the decision to submit the work for publication.

### Author contributions

Mohamed Abbas, Conceptualization, Data curation, Formal analysis, Funding acquisition, Investigation, Methodology, Visualization, Writing – original draft, Writing – review and editing; Anne Cori, Formal analysis, Investigation, Methodology, Writing – original draft, Writing – review and editing; Samuel Cordey, Funding acquisition, Investigation, Methodology, Writing – review and editing; Florian Laubscher, Formal analysis, Investigation, Methodology, Writing – review and editing; Tomás Robalo Nunes, Ashleigh Myall, Virginie Prendki, Anne Iten, Christophe E Graf, Investigation, Methodology, Writing – review and editing; Julien Salamun, Philippe Huber, Dina Zekry, Laure Vieux, Valérie Sauvan, Investigation, Writing – review and editing; Stephan Harbarth, Conceptualization, Formal analysis, Funding acquisition, Investigation, Methodology, Project administration, Supervision, Writing – original draft, Writing – review and editing

### Author ORCIDs

Mohamed Abbas ![ORCID] http://orcid.org/0000-0002-7265-1887
Samuel Cordey ![ORCID] http://orcid.org/0000-0002-2684-5680

### Ethics

Human subjects: The Ethics Committee of the Canton of Geneva (CCER), Switzerland, approved this study (CCER no. 2020-01330 and CCER no. 2020-00827). Written informed consent was obtained from HCWs. Written informed consent was not required for patients as data were generated in a context of mandatory surveillance.

### Decision letter and Author response

Decision letter https://doi.org/10.7554/eLife.76854.sa1
Author response https://doi.org/10.7554/eLife.76854.sa2

## Additional files

### Supplementary files
• Transparent reporting form

### Data availability
Due to small size of the various clusters, the raw clinical data will not be shared to safeguard anonymity of patients and healthcare workers. Processed data of the output of the model, which will comprise the posterior distribution of infectors, will be made available in an anonymized version. This will allow reproduction of the analyses looking at the proportion of healthcare workers among infectors, and the number of secondary infections. This data will not allow reconstruction of the transmission tree, which would require the raw data. The raw data in an anonymized format will be made available upon reasonable and justified request, subject to approval by the project's Senior Investigator. The genomic sequencing data have been submitted to the Genbank repository (GenBank accession numbers: ON209723-ON209871).

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

# Appendix 1

**Appendix 1—table 1.** Composition of baseline `outbreaker2` model and different sensitivity analyses.

| Scenario type | Onset of symptoms | Genetic data | Contact data | Short serial interval | Longer serial interval | Low outlier threshold | Medium outlier threshold | Default outlier threshold |
|---|---|---|---|---|---|---|---|---|
| Baseline scenario | X | X | X | X | | X | | |
| | | | | | | | | |
| Sensitivity analyses | | | | | | | | |
| 1. | X | X | | X | | X | | |
| 2. | X | X | X | | X | X | | |
| 3. | X | X | X* | X | | X | | |
| 4. | X | X | X† | X | | X | | |
| 5. | X | X | X | X | | | X | |
| 6. | X | X | X | X | | | | X |

*For this model, we used the HR data for healthcare worker presence (with some corrections).
†For this model, we assumed that patients were no longer infectious after the date of positive RT-PCR.

**Appendix 1—table 2.** Proportions of secondary infections (i.e., individual R) for each case type (patient$_{noso}$, HCW$_{outbreak}$, nd HCW$_{covid}$) in early (up to 9 April 2020) and late (as of 10 April 2020) phases of the study.
The p-values are for chi-squared tests on these proportions.

| | Patient$_{noso}$ | HCW$_{outbreak}$ | HCW$_{covid}$ | p-value (HCW$_{outbreak}$ vs. patient$_{noso}$) |
|---|---|---|---|---|
| Main analysis | | | | |
| ≥1 secondary transmission | | | | |
| Early phase | 0.68 (0.52–0.8) | 0.684 (0.526–0.789) | 0.442 (0.349–0.535) | 0.576 |
| Late phase | 0.471 (0.294–0.647) | 0.429 (0.286–0.714) | 0.417 (0.306–0.528) | 0.552 |
| ≥2 secondary transmissions | | | | |
| Early phase | 0.32 (0.2–0.44) | 0.316 (0.158–0.474) | 0.186 (0.093–0.256) | 0.459 |
| Late phase | 0.118 (0–0.294) | 0.286 (0–0.571) | 0.139 (0.056–0.222) | 0.807 |
| | | | | |
| Sensitivity analysis #1 (no assumptions about contacts) | | | | |
| ≥1 secondary transmission | | | | |
| Early phase | 0.64 (0.48–0.76) | 0.632 (0.474–0.789) | 0.442 (0.326–0.512) | 0.527 |
| Late phase | 0.529 (0.353–0.706) | 0.429 (0.143–0.714) | 0.389 (0.278–0.5) | 0.322 |
| ≥2 secondary transmissions | | | | |
| Early phase | 0.36 (0.24–0.52) | 0.263 (0.105–0.421) | 0.163 (0.093–0.256) | 0.190 |
| Late phase | 0.176 (0.059–0.353) | 0.143 (0–0.429) | 0.111 (0.028–0.195) | 0.527 |
| | | | | |
| Sensitivity analysis #2 (long serial interval) | | | | |

*Appendix 1—table 2 Continued on next page*

*Appendix 1—table 2 Continued*

| | Patient$_{noso}$ | HCW$_{outbreak}$ | HCW$_{covid}$ | p-value (HCW$_{outbreak}$ vs. patient$_{noso}$) |
|---|---|---|---|---|
| ≥1 secondary transmission | | | | |
| Early phase | 0.64 (0.52–0.76) | 0.684 (0.526–0.791) | 0.442 (0.349–0.535) | 0.669 |
| Late phase | 0.471 (0.294–0.647) | 0.429 (0.282–0.714) | 0.389 (0.278–0.5) | 0.561 |
| ≥2 secondary transmissions | | | | |
| Early phase | 0.36 (0.2–0.48) | 0.316 (0.158–0.474) | 0.186 (0.116–0.279) | 0.445 |
| Late phase | 0.118 (0–0.294) | 0.286 (0–0.571) | 0.139 (0.056–0.222) | 0.790 |
| | | | | |
| **Sensitivity analysis #3 (calibrating contacts based on assumptions on infectiousness)** | | | | |
| ≥1 secondary transmission | | | | |
| Early phase | 0.68 (0.56–0.8) | 0.684 (0.526–0.789) | 0.442 (0.349–0.535) | 0.433 |
| Late phase | 0.471 (0.235–0.647) | 0.286 (0.143–0.571) | 0.361 (0.25–0.472) | 0.249 |
| ≥2 secondary transmissions | | | | |
| Early phase | 0.4 (0.24–0.52) | 0.368 (0.211–0.474) | 0.186 (0.116–0.256) | 0.360 |
| Late phase | 0.118 (0–0.294) | 0.143 (0–0.429) | 0.083 (0.028–0.167) | 0.702 |
| | | | | |
| **Sensitivity analysis #4 (patients no longer infectious after date of swab)** | | | | |
| ≥1 secondary transmission | | | | |
| Early phase | 0.64 (0.48–0.8) | 0.684 (0.526–0.842) | 0.465 (0.349–0.535) | 0.627 |
| Late phase | 0.471 (0.294–0.706) | 0.429 (0.286–0.714) | 0.417 (0.306–0.528) | 0.522 |
| ≥2 secondary transmissions | | | | |
| Early phase | 0.32 (0.16–0.44) | 0.316 (0.158–0.475) | 0.186 (0.116–0.279) | 0.541 |
| Late phase | 0.118 (0–0.294) | 0.286 (0–0.571) | 0.111 (0.028–0.194) | 0.781 |
| | | | | |
| **Sensitivity analysis #5 (higher value for outlier threshold)** | | | | |
| ≥1 secondary transmission | | | | |
| Early phase | 0.64 (0.48–0.721) | 0.684 (0.526–0.789) | 0.419 (0.326–0.512) | 0.683 |
| Late phase | 0.471 (0.294–0.647) | 0.429 (0.143–0.714) | 0.417 (0.306–0.528) | 0.542 |
| ≥2 secondary transmissions | | | | |
| Early phase | 0.32 (0.16–0.48) | 0.316 (0.158–0.474) | 0.186 (0.093–0.256) | 0.407 |
| Late phase | 0.118 (0–0.294) | 0.286 (0–0.571) | 0.111 (0.028–0.194) | 0.774 |
| | | | | |
| **Sensitivity analysis #6 (default value for outlier threshold)** | | | | |
| ≥1 secondary transmission | | | | |
| Early phase | 0.64 (0.48–0.76) | 0.684 (0.526–0.789) | 0.442 (0.349–0.512) | 0.682 |

*Appendix 1—table 2 Continued on next page*

Appendix 1—table 2 Continued

|  | Patient$_{noso}$ | HCW$_{outbreak}$ | HCW$_{covid}$ | p-value (HCW$_{outbreak}$ vs. patient$_{noso}$) |
| --- | --- | --- | --- | --- |
| Late phase | 0.471 (0.294–0.647) | 0.429 (0.286–0.714) | 0.417 (0.306–0.528) | 0.522 |
| ≥2 secondary transmissions |  |  |  |  |
| Early phase | 0.32 (0.2–0.48) | 0.316 (0.158–0.474) | 0.186 (0.116–0.279) | 0.424 |
| Late phase | 0.118 (0–0.294) | 0.286 (0–0.571) | 0.139 (0.056–0.222) | 0.790 |

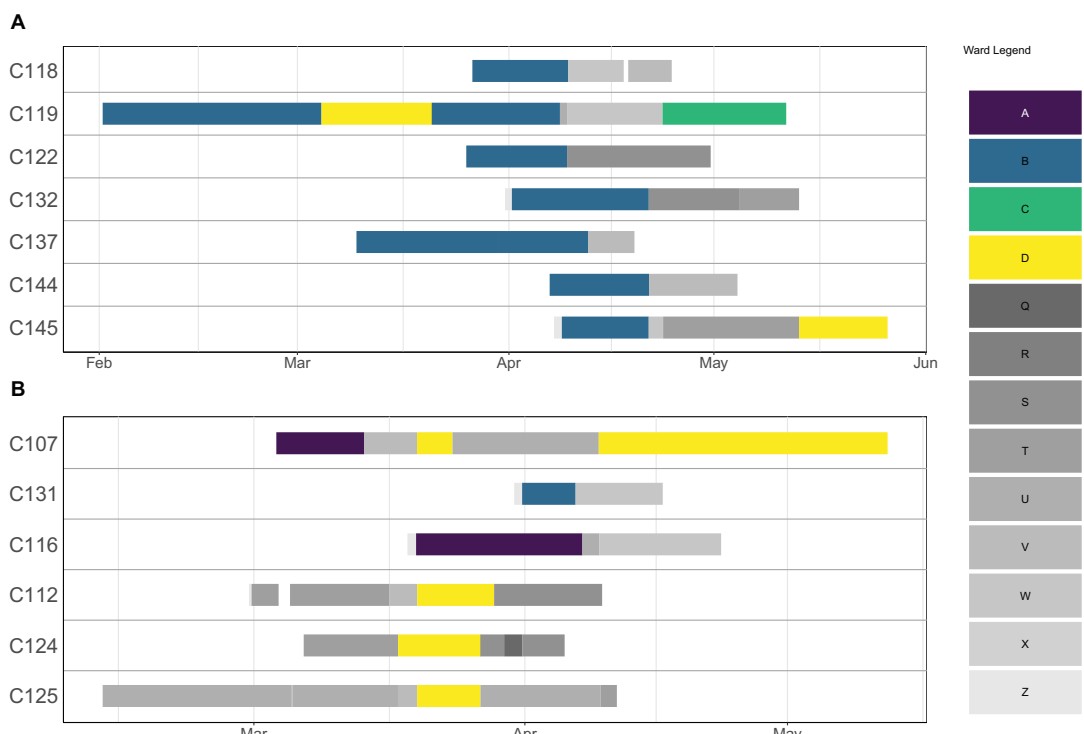

**Appendix 1—figure 1.** Ward movements for patients involved in a cluster. Each row corresponds to a patient, and the solid lines indicate hospitalisation dates. The lines are coloured according to which ward a patient was in on a particular day. Outbreak wards (A–D) are coloured differently from non-outbreak wards (Q–Z).

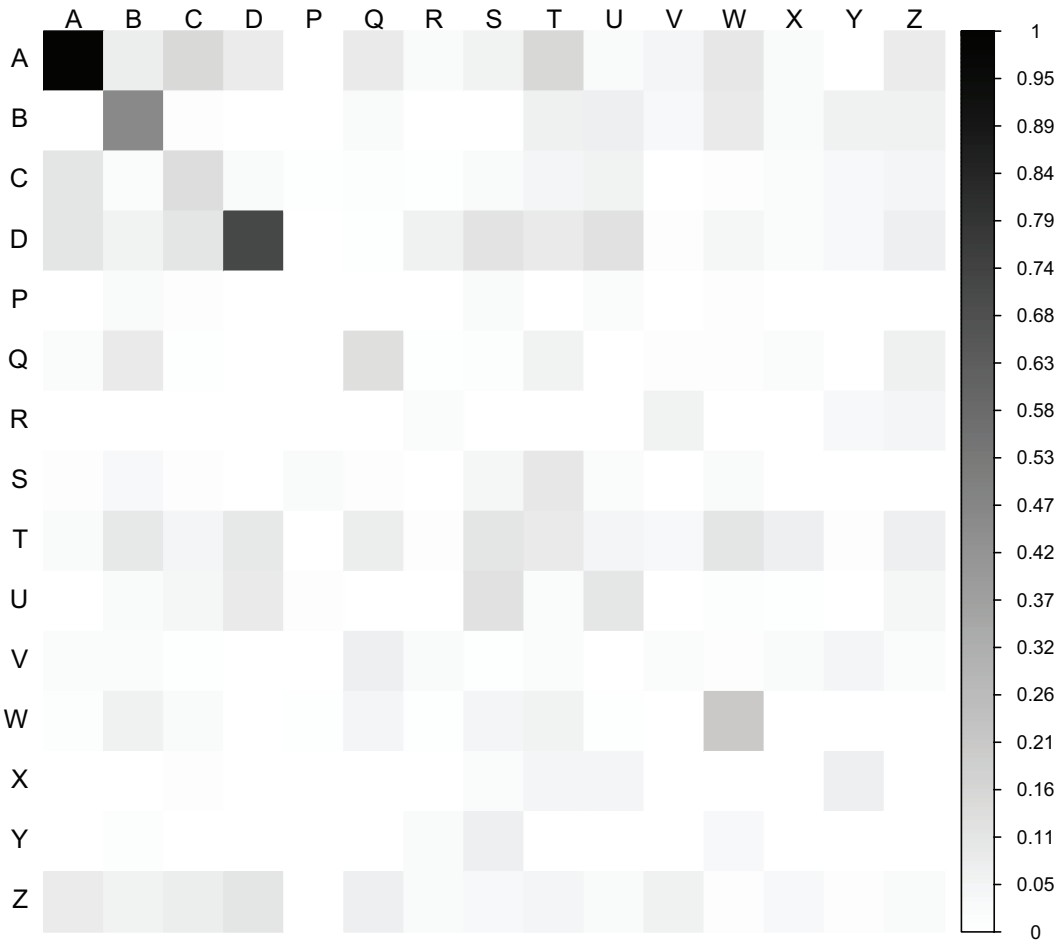

**Appendix 1—figure 2.** Ward-to-ward transmission matrix. The matrix indicates the sum of transmission events across all posterior trees from cases in 'infector' wards (vertical axis) to cases in 'infectee' wards (horizontal axis). The degree of shading is proportional to the estimated posterior number of transmissions for each ward-to-ward pair. Outbreak wards: A–D; non-outbreak wards: P–Z (Z is 'all wards').

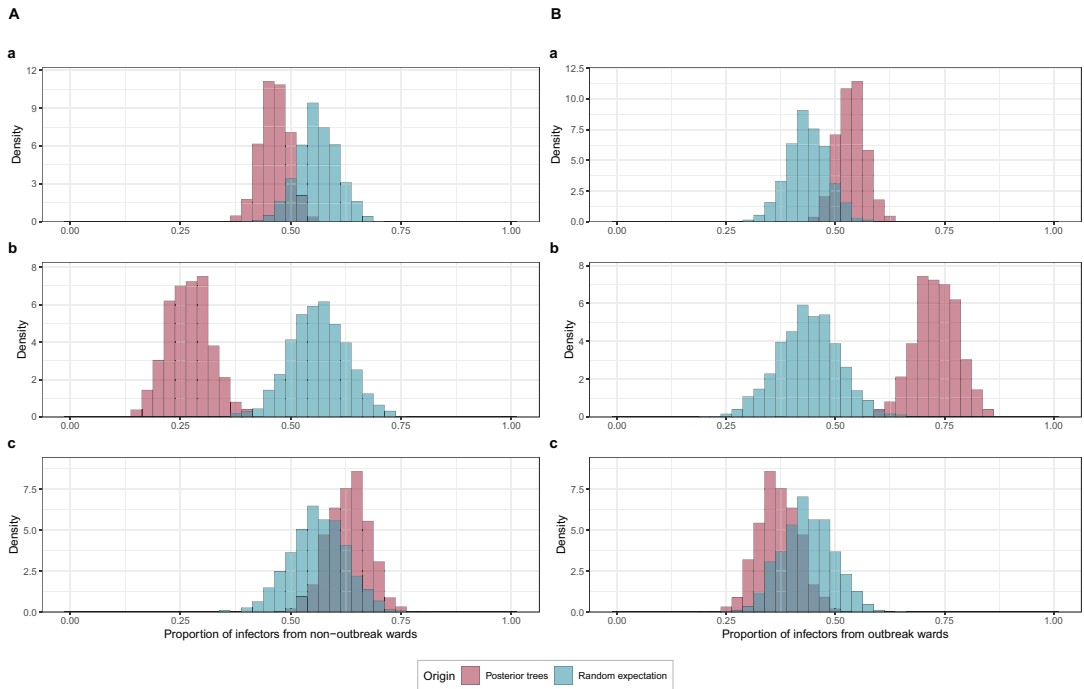

**Appendix 1—figure 3.** Proportions of transmissions attributed to (**A**) outbreak ($f_{outbreak-ward}$) and (**B**) non-outbreak ($f_{non-outbreak-ward}$) wards. The blue histograms indicate the expected random distributions of $f_{ward}$, given the proportion of HCWs amongst cases. The red histograms show the observed distribution of $f_{ward}$, across 1000 transmission trees reconstructed by `outbreaker2`. (**A**). All wards. (**B**) Transmission to outbreak wards only. (**C**) Transmission to non-outbreak wards only.

## Infection prevention and control measures during the first pandemic wave

A multidisciplinary Covid-19 response unit was formed in the Geriatric hospital, with daily meetings and involvement of the IPC team. Direct coaching of HCWs working in Covid-19 wards by the IPC practitioner was undertaken once a ward had been attributed as such. The IPC practitioner assisted front-line HCWs in streamlining tasks.

HCWs caring for Covid-19 patients applied 'contact' and 'droplet' precautions, in line with Swissnoso and FOPH guidelines. Universal masking of all front-line HCWs was implemented on 11 March 2020. From 1 April 2020, the use of ocular protection (eye shields) was encouraged for contact with all Covid-19 patients, and masking of HCWs in non-clinical areas (e.g., offices) was recommended.

RT-PCR screening of SARS-CoV-2 on admission for all patients, even if asymptomatic or without clinical suspicion of Covid-19, started on 1 April 2020. From 7 April 2020, weekly screening surveys were performed in non-Covid wards, until 30 May 2020.

As of 11 April 2020 non-Covid wards were closed to new admissions, and room occupancy was decreased (e.g., four-bed rooms were limited to three patients).

Due to shortages in PPE, HCWs were instructed to wear surgical masks for 4 hr continuously (and 8 hr if not humid), and N95 respirators (FFP2 masks) for as long as possible. Gowns were used for >1 patient in the same room, except in the case of patients carrying multidrug-resistant bacteria.

HCWs from outbreak wards were encouraged to undergo PCR testing on nasopharyngeal swabs, even if asymptomatic between 9 and 16 April 2020. Including tests that were performed since mid-March, a total of 83 out of 124 eligible HCWs (67%) in the four outbreak wards underwent testing.

We split the outbreak into two phases, up to 9 April 2020 and after 9 April 2020 for several reasons. First, many preventive measures were implemented around that date (e.g., weekly screening of patients on 7 April enhanced testing of HCWs on 9 April, decrease in bed-occupancy and closures on 11 April). Second, approximately half of the cases occurred in each phase of the outbreak. Third,

the overall trend in the epidemic curve can be interpreted as increasing in the first phase, and decreasing in the second phase.

## Ward architecture and room sizes in the geriatric hospital

The architectural layout of the hospital means that there are two wings per floor, each with a central corridor, and two wards per wing (without separation). On either side of the central corridor are patient rooms and/or offices (nursing, medical, etc.). Therefore, the wards themselves are crowded areas, and it is sufficient for a patient to step outside their room in the corridor to potentially come in contact with another patient.

The mean room sizes (for rooms that were shared) were small, being 23.1 $m^2$ (standard deviation 6.5 $m^2$) and 27.8 $m^2$ (standard deviation 0.2 $m^2$) for two bed and three bed rooms, respectively.

## Microbiological methods

### Amplicon-based high-throughput sequencing analysis

All nasopharyngeal swabs (NPS) tested positive for SARS-CoV-2 by RT-PCR (Cobas 6800 SARS-CoV2 RT-PCR), the Charite or the BD SARS-CoV2 reagent kit for BD Max system assays, and for which sufficient volume remained, were selected for whole-genome sequencing analysis.

NPS were sequenced with an amplicon-based sequencing method. Thus, nucleic acids were extracted using the NucliSENS easyMAG (bioMérieux, Geneva, Switzerland) and then sequenced using an updated version of the nCoV-2019 sequencing protocol (https://www.protocols.io/view/ncov-2019-sequencing-protocol-bbmuik6w) (Microsynth, Balgach, Switzerland) on a MiSeq instrument (Illumina) with a 2×250 bp protocol.

Duplicate reads were removed using cd-hit (v4.6.8). Low-quality and adapter sequences were trimmed out using Trimmomatic (v0.33). Reads were then mapped against the reference sequence MN908947 using snap-aligner (v1.0beta.18). Consensus for sequences with at least 10-fold coverage were then generated using custom script.

### Phylogenetic analysis

Sequence alignment was performed with MUSCLE (v3.8.31). Evolutionary analyses were conducted in MEGA X (*Kumar et al., 2018*) using the Maximum Likelihood method and Tamura three-parameter model (*Tamura, 1992*). The tree includes also all SARS-CoV-2 complete genomes sequenced by our laboratory and submitted to GISAID from respiratory samples from COVID-19 positive patients presenting to our institution or other medical centres in Geneva, Switzerland, during the same period.

## Statistical analyses

### Implementation of the `outbreaker2` models

We combined epidemiologic and genetic data using the `outbreaker2` package in the R software, which has been used successfully in the reconstruction of the 2003 SARS-CoV-1 outbreak in Singapore (*Jombart et al., 2014*; *Campbell et al., 2018*) and a nosocomial outbreak of SARS-CoV-2 in a rehabilitation clinic (*Abbas et al., 2021a*). The model uses a Bayesian framework, which combines information on the generation time (time between infections in an infector/infectee pair), with a model of sequence evolution to probabilistically reconstruct the transmission tree.

As dates of onset of symptoms are known, and because dates of infection (i.e., acquisition) are not known with certainty, we imputed serial intervals based on estimates from the work by *Ali et al., 2020*, who showed that the serial interval decreased from the early stages of the pandemic due to improved control using non-pharmaceutical measures. For the primary analysis we used a short serial interval (mean 3.0, standard deviation [SD] 4.1), under the assumption of swift isolation of patients following onset of symptoms. We also performed a sensitivity analysis using a longer serial interval (mean 5.2 days, SD 4.7) to allow for potentially slower isolation of symptomatic patients. We used the incubation period as estimated by *Bi et al., 2020*, which follows a lognormal distribution with parameters mu of 1.57 and sigma 0.65 (corresponding to a mean of 5.95 days and SD 4.31). Where dates of onset were unavailable, we imputed them by using the median of the difference between symptom onset and date of swab.

Imported cases are detected by `outbreaker2` during a preliminary run of the model, where cases are removed in turn (with a leave one out approach) and the 'global influence' of each case

is measures as the extent to which removing it affects the genetic log-likelihood. Cases with high global influence (whose removal dramatically affects the genetic likelihood) are considered to be genetic outliers, and classified as imported cases. This approach is described in more detail in the original `outbreaker2` manuscript (*Jombart et al., 2014*). By default, a case is determined as being imported if its global influence is five times the average global influence. While this approach has excellent specificity, because of the limited genetic diversity of SARS-CoV-2, it may lack in sensitivity. We therefore ran the model over a range of lower thresholds (from 2 to 5 times the average global influence). We selected for our main model a threshold of 2, and present one-way sensitivity analyses with a threshold of 3 and the default (9).

The `outbreaker2` package is designed to use contact tracing data to inform who infected whom. These data were unavailable for our outbreak, but we made a series of assumptions to generate a matrix of possible contacts between cases. Initially, we aimed to construct this matrix using dates of presence in the hospital/ward for patients based on administrative data, and based on human resources shift rota for HCWs. However, we identified potential inconsistencies in the latter, in particular stemming from multiple changes as a result of many HCWs self-isolating due to possible or confirmed COVID-19. We therefore used these data in a sensitivity analysis, but for our main analysis, we reverted to a simpler set of assumptions to build our contact matrix. In our main analysis, we assumed that HCWs were present in the hospital every day until the date of their first positive swab, included. We further assumed that patients only interacted with patients in their own ward. We assumed that all HCWs were able to infect each other, but HCWs could have contact with patients only in the wards they were attributed to; HCWs such as physical therapists or doctors who worked across multiple wards could have significant contact with all HCWs and patients. Under these assumptions, we are likely to capture many contacts which did not happen, and it is possible that we miss a few contacts which in fact did happen. However `outbreaker2` does account for imperfect sensitivity and specificity of contact data, the levels of which are estimated as part of the model. To account for this uncertainty in potential contact patterns, we ran two sensitivity analyses where we input different contact data in the model.

## Sensitivity analyses

We conducted a number of different analyses, including a base scenario and several (n=5) one-way sensitivity analyses (*Appendix 1—table 1*):

- Sensitivity analysis 1:
  - We did not make any assumptions about contact patterns, and therefore all cases in the outbreak could potentially infect all other cases.
- Sensitivity analysis 2:
  - We used a longer serial interval (mean 5.2 days, SD 4.7) to allow for potentially slower isolation of symptomatic patients.
- Sensitivity analysis 3:
  - We used the HCW shift data from the human resources department, with minor corrections (removing HCWs that were mislabelled as 'present' after date of positive RT-PCR). For both patients and HCWs we categorised days of 'susceptibility' (fifth percentile of the cumulative incubation period from *Bi et al., 2020*) and days of 'infectiousness' (2 days before symptom onset based on the study by *He et al., 2020*). The last day of 'infectiousness' was the date of swab for HCWs.
- Sensitivity analysis 4:
  - We assumed that isolation precautions prescribed for patients on date of positive RT-PCR were effective, and that from that date patients were no longer infectious.
- Sensitivity analysis 5:
  - We used a higher threshold of 3 for the determination of imported cases.
- Sensitivity analysis 6:
  - We used the default threshold (5) used by `outbreaker2` to detect imported cases.

For each model, we used a uniform prior between 0.55 and 1 for the reporting probability (pi). Indeed, we had a comprehensive screening and testing strategy, including of asymptomatic cases, and are

therefore confident that we captured a near-total proportion of cases. We obtained sequences for 82% of all identified cases, and these are the cases used in the model. The lower bound of the prior for 'pi' thus allows us to have missed six cases in addition to the 14 that were not sequenced. Posterior estimates for 'pi' were visually compared to our prior choice to assess the validity of this assumption.

We allowed for a maximum of two unobserved cases on a transmission chain between any two observed cases (maximum 'kappa' of 3 in `outbreaker2`). This allows for identification of missed cases.

We used the default priors for the mutation rate (mu) for all models (uninformative exponential prior with mean 1), and, where relevant, those for non-infectious contact rate (lambda) and contact reporting coverage (eps), which were uniform on [0, 1] (*Campbell et al., 2018*; *Campbell et al., 2019*). We used the default likelihoods for all models, except for the model without contact data where this was disabled (*Campbell et al., 2018*; *Campbell et al., 2019*).

We ran each `outbreaker2` model over 2,000,000 iterations of the MCMC (500,000 for model without contact data), with a thinning of 1 in 2000 (1 in 500 for model without contact data), in order to obtain a sample of 1000 posterior parameter sets, after a burn-in of 2000 iterations (500 for model with contact data). Each of these parameter sets corresponds to a posterior transmission tree. Convergence was assessed visually as well as through the Gelman-Rubin convergence diagnostic (using the gelman.diag function in the R package coda v0.19-4) (*Gelman and Rubin, 1992*), concluding that the chains converged appropriately if the upper limit of the confidence interval was <1.1.

To assess the role of each type of case in transmission by estimating the proportion of infections attributed to the type of case ($f_{case}$), we compared the posterior distribution of direct infections caused by each infector type to that obtained by a matching number of random draws of the infector types (expected proportion of infections), drawn according to the prevalence of each type among cases. We concluded that $f_{case}$ was significantly higher than expected by chance if at least 95% of the posterior samples had higher proportions of cases infected by an infector type than that obtained by the random draws. The corresponding p-values were calculated as 1 minus the proportion of posterior samples with values higher than random draws. This was done for all infectees (i.e., whole outbreak), as well as for each type of infectee.

We evaluated the differences in the distribution of secondary cases across case types in the early (up to 9 April 2020) and later phases (as of 10 April 2020) of the study period using a chi-squared test (for proportions of cases with no secondary transmissions [non-transmitters] and of cases with ≥2 secondary transmissions [high transmitters]).

