## [Editor Report]

Congratulations on this useful and technically impressive paper demonstrating that phylogenetic and epidemiologic data can be used in a retrospective cohort to reconstruct that chain of events in terms of case importation into a high risk geriatrics ward. The conclusion that HCW (healthcare worker) transmission in non-COVID wards was particularly important is critical for hospital epidemiologists. The methodology advances will hopefully push the field forward in terms of tracking outbreaks in various settings.

---

## [Decision Letter]

**Decision letter after peer review:**

Thank you for submitting your article "Reconstruction of transmission chains of SARS-CoV-2 amidst multiple outbreaks in a geriatric acute-care hospital: a combined retrospective epidemiological and genomic study" for consideration by *eLife*. Your article has been reviewed by 3 peer reviewers, and the evaluation has been overseen by a Reviewing Editor and Wendy Garrett as the Senior Editor. The following individuals involved in the review of your submission have agreed to reveal their identity: Lulla Opatowski (Reviewer #1); Thi Pham (Reviewer #3).

Essential revisions:

1) Clarify and improve Figure 3.

2) If possible, add any available details about transmissions between ward-sharing individuals in terms of distance between rooms and beds.

3) Add discussion about which results may or may not be generalizable to other health care settings, as well as the issue of misclassifying the infection source.

*Reviewer #1 (Recommendations for the authors):*

– Abstract line 63, sentence starting with "the proportion of infectors…" should be clarified.

– Because the analysed data provides from the first pandemic wave, several variables may have changed substantially over the studied period. In particular, as described in the methods section, sampling policy varied over time. Sampling was generally achieved differently in HCWs and patients. Were these two aspects accounted for in the statistical model? How does that affect uncertainty on estimates?

– What is the level of certainty for provided dates of symptom onset? Were they extracted from medical records or collected in real-time? To what extent does the reconstruction of the transmission chain rely on this variable and was uncertainty considered on that variable?

– Do all patients share rooms, and with how many others? What is this 44% as a proportion of those who do?

– There is a long-tailed distribution of infection by each individual (Figure 4). How much do heavy spreaders influence the results e.g. would the results from Figure 5 all hold if individual superspreader events were removed from the dataset?

– When calculating expectation in Figure 5, this is calculated by the prevalence of Covid-19 in each type of individual on a given ward. In our understanding, the ward was not considered here. Could the authors justify that choice? Would it be feasible to calculate an expectation which took into account the higher risk of transmission when sharing a room, and use this as the comparator against observed infections?

– Page 11 line 231, the referenced figure doesn't indicate these probabilities. What is the source?

– How was the date of April 9 fixed to define a cut-off between the early phase and late phase? Is it just based on the peak incidence? Or due to a change in screening practice?

– The large outbreak size makes figure 3 quite hard to interpret for the reader. Would it be possible to improve it possibly by zooming on some specific cases or wards, and adding global statistics informing on the precision/quality of the inference of transmission routes?

– How does the number of secondary infections relate to the population's sizes (i.e. at-risk patients and HCWs from the studied wards). Is that relationship expected?

– We may have missed this: (apologies if so): were HCWs cohorted by wards or were there HCWs shared among wards? How may that affect the results here?

*Reviewer #2 (Recommendations for the authors):*

This paper addresses an important question about the dynamics of SARS-CoV-2 in a geriatric acute-care hospital. This reviewer's comments address primarily manuscript clarity.

General Comments:

Patients should not be equated with their conditions. Please replace Covid-19 and non-Covid 19 patients with patients who have Covid-19 and patients who do not have Covid-19 throughout the manuscript.

Compare takes "with" not "to".

The manuscript includes many "there is," "there are," "it is," xxx. These statements are in passive voice and are usually wordier and more convoluted than a similar statement in active voice. This reviewer suggests revising most or all of these statements. For example, the following sentence could be revised as shown.

"The reservoir of SARS-CoV-2 in healthcare environments may contribute to amplifying the pandemic [5], and as such, it is important to understand transmission dynamics in these settings."

"Because the reservoir of SARS-CoV-2 in healthcare environments may contribute to amplifying the pandemic [5], we need to better understand transmission dynamics in these settings."

Statements like "in order to" and "as well as" have extra words that don't add to the meaning. They can be shortened to "to" and "and".

Specific Comments

Line 67: Is the word "unexpectedly" needed? This reviewer was confused by that statement. I would assume HCWs knowing that they were caring for patients with Covid-19 might be more in adherence with PPE use and hand hygiene than those on non-Covid-19 units.

Lines 93-95: The meaning of "in nosocomial Covid-19" is vague and should be clarified. Does this mean HCWs' role in transmitting SARS-CoV-2 within healthcare facilities is complex?

Does it mean HCWs role in the epidemiology of nosocomial SARS-CoV-2 infection is

complex? This reviewer does not like the word "victim" (although the alteration is nice) in this setting. This reviewer thinks the last part of this sentence will work better if revised to: "as HCWs can acquire SARS-CoV-2 in the community or from their peers and patients and they can transmit SARS-CoV-2 to peers and patients [7, 8].

Methods

The authors switch frequently between active and passive voice. This reviewer suggests picking one and sticking with it. This reviewer prefers an active voice for clarity and brevity.

Line 162: This reviewer does not understand what the authors mean by "trajectories" in this sentence.

Lines 165-6: Revise to something like: "conservative in that it accounts for non-infectious contacts and incomplete reporting of infectious contacts and it estimates the proportion of the total contacts in these categories."

Lines 18-91: Suggested revision. A post hoc epidemiological analysis of the 53 patients found that 4 of these patients likely acquired Covid-19 in the community (CA-Covid-19). The 49 nosocomial cases represented 20.2% (49/242) of all patients with COVID-19, and 81.7% (49/60) of nosocomial Covid-19 cases identified in the Geriatric Hospital. The ward-level attack rates ranged from 10 to 19% among patients and 21% of all HCWs in the geriatric..."

Line 203: This reviewer does not understand this sentence given that the verb appears to be "shows".

Lines: 273-5: Suggest revising to something like: "Fourth, we identified multiple importation events that led to a substantial number of secondary cases or clusters. Most transmission events were related to HCWs, but one (?? or some) were related to a patient with community-acquired Covid-19." Note: the authors did not make clear how many transmissions were related to this patient.

Lines 282-292: Suggest revising to something like: "HCWs' behavior may have affected transmission. Indeed, Ottolenghi et al., found that HCWs caring for Covid-19 patients were concerned about becoming infected while caring for patients, and therefore may apply IPC measures more rigorously than when caring for patients who do not have Covid-19 [23]. HCWs working in non-Covid wards may not have felt threatened by Covid-19 patients on other wards. HCWs in non-Covid wards also may have underestimated the transmission risk from their peers to a greater extent than those working in Covid wards, and thus did not maintain physical distancing well. In addition, we found a higher mean duration (2.9 days) of presenteeism despite symptoms compatible with Covid-19 among HCWs working in non-Covid wards than for those working in Covid wards (1.6 days), which gives credence to the abovementioned possible explanations for the different transmission patters. Other factors (e.g., work culture, baseline IPC practices) also may have affected transmission patterns.

This reviewer wonders if the following sentence is necessary? "HCWs working in non-Covid wards may not have felt threatened by Covid-19 patients on other wards."

Line 294: This author is wondering how the same ward is close proximity when patients are not in the same room. How would the patients have contact with each other?

Lines 294-309: Suggested revisions: We cannot exclude transmission from a "point-source" or via an HCW's contaminated hands (i.e., an unidentified HCW who transmitted SARS-CoV-2 to multiple patients in the same ward). To date, we have little evidence to suggest that this is the case; indeed, the transmission patterns were robust to changes in the model assumptions. Mathematical models have suggested that single-room isolation of suspected cases could potentially reduce the incidence of nosocomial SARS-CoV-2 transmission by up to 35% [24]. In the outbreaks we describe, symptomatic patients were identified promptly, with a median delay of 0 days between symptom onset and their first positive test. However, these precautions may not be sufficient as patients may transmit the virus when they are pre-symptomatic [25]. Thus, infection prevention teams may need to identify patients at high risk of developing nosocomial Covid-19 [26] if single rooms are not available for all exposed patients (e.g., in cases of overcrowding). For example, Mo et al., found that exposure to community-acquired cases who were identified and segregated or cohorted was associated with half the risk of infection compared with exposure to hospital-acquired cases or HCWs who may be asymptomatic [27]. One possible explanation for this finding is that patients with CA-Covid may have passed the peak of infectiousness when they are admitted, whereas patients with HA-Covid cases have frequent unprotected contact with HCWs and other patients during their period of peak infectiousness.

Line 310: Suggested edit: The current evidence does not support the use of real-time genomics for control of SARS-CoV-2 nosocomial outbreaks [28].

Lines 324-328: Suggested edits: Despite these strengths, our study had some limitations. First, we included only one sequence from a CA-Covid. However, the method we used to reconstruct who infected who is able to cope with and identify missing intermediate cases, which allowed us to estimate that our sample included the overwhelming majority of cases (91.5%). In addition, we performed these investigations during the first pandemic wave in a susceptible population...."

Line 334: Replace "in the case of" with "during".

Lines 336-338: The main point of the conclusion is not clear. What does it mean to take into account the complex interplay between HCWs in dedicated Covid-19 wards vs non-Covid wards? The way this is worded makes it sound like the HCW on the two different kinds of wards are interacting, which likely is not what the authors intended.

Table 1: This reviewer doesn't understand what is meant by "Onset of symptoms before swab date, n (%)". Is this the number of people who were symptomatic before they were tested?

Table 2: Should HCWs be written out in the title? Readers in the US will likely not know what a logopedist is. We will call this person a speech therapist. Perhaps the authors could include a footnote defining this term. Did the authors address the longer time from onset of symptoms to testing for HCWs on non-Covid wards compared with those on Covid wards in the results and discussion?

Table 3: Dose "Secondary onward transmission" refer to transmission from the case in the first column? Consider putting NA (not applicable) in the blank cells.

Line 515: Suggested edit: "Histograms displaying the distributions of secondary cases...."

Line 525: Is something missing between "... patient community and "proportion"?

*Reviewer #3 (Recommendations for the authors):*

Methods:

The authors present the IPC measures on page 6 in the main text and in the supplementary material. However, there seems to be no information on how HCWs adhered to the recommendations that were not mandatory and whether adherence differed between wards. This is probably because the authors did not have any information on that. This is a limitation and should be explained more explicitly or highlighted in the discussion.

It's not clear to me whether the HCWs only worked in either COVID or non-COVID wards during the study period. Please clarify in the Methods section.

On page 7, line 139: The authors state that they substituted the missing date of symptom onsets with the median difference between symptom onset date and swab date? The epidemic curve in Figure 1 shows the incidence on the y-axis but does the x-axis (date of symptom onset) make sense if not all of the cases were symptomatic?

Page 8, line 169 and Figure 4: How was the epidemic phase determined?

Results:

At the beginning of the Results section, the authors write that post hoc epidemiological analysis showed that were likely 4 patients that had a community-acquired infection. What exactly did the post-hoc epidemiological analysis entail? Later on (page 10, lines 206ff), they present their analysis of the imported cases. How are these related?

Discussion:

Could the authors discuss whether misclassifications (either of community- or hospital-acquired cases) even after the analysis is still possible and what impact it could have on their results?

Supplement

Page 5: Could the authors elaborate on "global influence" for imported cases?

---

## [Author Response]

Essential revisions:1) Clarify and improve Figure 3.

We agree with Reviewer #1 that the initial Figure 3 was difficult to read. We have now created an interactive chart where one can hover over points and zoom in selected areas. We have also removed the colours, which did not have a particular meaning. Following a suggestion from Reviewer #1, we have decided to move this new interactive figure as Figure 1 – Supplementary figure 1, and have replaced it with a more informative figure demonstrating the extent to which our model is able to infer of transmission routes (i.e., showing the distribution of the posterior support of ancestries based on different criteria (individual, case type, ward), see new Figure 3).

Legend for New Figure 3: Distribution of posterior support of maximum posterior ancestry for all cases, according to identity of A individual ancestor, B ancestor’s case type (i.e., “HCW_covid_”, “HCW_outbreak_”, “patient_noso_”, “patient_community_”), C ancestor’s ward, and D ancestor’s ward type (i.e., “outbreak ward”, “non-outbreak ward”).

2) If possible, add any available details about transmissions between ward-sharing individuals in terms of distance between rooms and beds.

We have added information about room sizes in the Appendix (page 3): “Indeed, the room sizes (for rooms that were shared) were small, being 23.1 m^2^ (standard deviation 6.5 m^2^) and 27.8 m^2^ (standard deviation 0.2 m^2^) for 2-bed and 3-bed rooms, respectively.” Unfortunately, we do not have precise measurements on distance between beds, but in general, beds next to each other were separated by approximately 1.5 – 2 metres.

When responding to a related comment from from Reviewer #1, we noticed an error in the pre-processing stage of contact data (n = 839 contacts [4%] were included that should not have been). We therefore re-ran all our analyses after correcting this error. This has led to minor changes in our numeric results but has not altered our main findings and conclusions. We apologise for this inconvenience, and are grateful to have had the opportunity to double-check our work as a result of the high-quality in-depth reviews.

3) Add discussion about which results may or may not be generalizable to other health care settings, as well as the issue of misclassifying the infection source.

We have added discussion points on the lack of generalisability of our results: “In addition, we performed these investigations during the first pandemic wave with a wild-type variant of SARS-CoV-2 in a susceptible population. For these reasons, the results may no longer be applicable in settings with high vaccination coverage and/or substantial natural immunity, or in later stages of the pandemic with different variants, also due to accrued experience in managing and preventing outbreaks. Nevertheless, the lessons learned may be useful in a large number of countries with slow vaccine roll-out due to vaccine hesitancy, particularly among HCWs where there is no vaccine mandate, or unequal access to vaccine supplies [33]. Furthermore, nosocomial outbreaks of SARS-CoV-2 still occur despite high vaccination coverage [34, 35]. Also, these valuable lessons may be applicable for nosocomial outbreak control in the case of future pandemics due to respiratory viruses with characteristics similar to SARS-CoV-2.” Part in italics was already in the text.

Misclassifications may still occur, and it is even part of the method (outbreaker2) to quantify the uncertainty in determining the ancestry, as it operates within a Bayesian framework. Indeed, in most cases (c.f. new Figure 3) there are very few patients for whom the ancestry is identified with 100% posterior probability. For example, the model predicted that patients C107 and C115 were most likely imported cases with posterior probabilities of 100% and 57.5%, respectively. The predicted dates of infection were in the range March 14-22 for case C107 and March 8-24 for case C115, which was after their admission dates (March 3 and March 18, respectively). The probability that infection occurred on or after date of admission for case C115 was 81%. Therefore the model acknowledges the uncertainty in the classification of these cases as imported. We have modified the section on imported cases (under the heading “imported cases”), and added a sentence after referencing the new Figure 3 to highlight the uncertainty in the model.

Finally, the fact that a case has a probability of being “imported” does not preclude the fact that it is still nosocomial cases, simply that their infector was not identified in this outbreak c.f. update in the manuscript (lines 241-259).

Reviewer #1 (Recommendations for the authors):– Abstract line 63, sentence starting with "the proportion of infectors…" should be clarified.

We agree that the sentence was unclear, and have modified it to “The proportion of infectors being HCW_covid_ was as expected as random”. We hope that it is now clearer.

– Because the analysed data provides from the first pandemic wave, several variables may have changed substantially over the studied period. In particular, as described in the methods section, sampling policy varied over time. Sampling was generally achieved differently in HCWs and patients. Were these two aspects accounted for in the statistical model? How does that affect uncertainty on estimates?

The reviewer raises an interesting and important point. Screening policies were indeed different between patients and HCWs. For patients, weekly screening surveys were performed in non-Covid wards from April 07, 2020 until May 30, 2020 (Appendix, page 3). In contrast, HCWs from outbreak wards were encouraged to undergo PCR testing, even if asymptomatic, between April 09 and April 16, 2020. These aspects were not explicitly accounted for in the statistical model. However, we are confident that the intensity of screening we performed during that period, led to a good case ascertainment. We note that the method we rely on, outbreaker2, does allow for undetected cases. It assumes a constant case reporting probability, which is estimated. In our study, this reporting probability estimated to be 91.5% (95% credible interval 91.2-91.9%), which confirms that case ascertainment was very high.

Outbreaker2 also estimates, for each observed case, the probability that one or more cases were unobserved in the transmission chain leading to their closest observed ancestry. We compared the posterior distribution of this number of “missed” generations between the early and late phases of the epidemic (i.e. before and after April 09, 2020). We have plotted the distribution of the number of missed generations across posterior trees, stratified by the phase. This suggests very little difference in the distribution of missed generations between the two phases. We have included this figure as Figure 3 – Supplementary Figure 2.

– What is the level of certainty for provided dates of symptom onset? Were they extracted from medical records or collected in real-time? To what extent does the reconstruction of the transmission chain rely on this variable and was uncertainty considered on that variable?

The dates of symptom onset were collected prospectively in real-time through several sources, minimising risk of bias. For patients, data were generated by the prospective surveillance of hospitalised patients with Covid-19 mandated by the Swiss Federal Office of Public Health, and for HCWs from both the hospital’s department of Occupational Health and from the prospective cohort study of outpatient Covid-19 testing in the hospital. We have clarified this in the “Data sources” section: “Dates of symptom onset were available for both patients and HCWs each source, respectively.”

The date of symptom onset is a main input of the model, as it is the starting point from which the date of infection is inferred. Although we did not directly account for potential uncertainty in the date of onset, our model considers whole distributions for the serial interval and incubation period which would allow absorbing a small level of uncertainty in onset dates, assuming such uncertainty is similar to those in studies which estimated the serial interval and incubation period.

– Do all patients share rooms, and with how many others? What is this 44% as a proportion of those who do?

Out of all 43 patients, there were 5 patients (9.4%) who did not share a room with another Covid patient; this number increases to 11 patients (20.7%) if a “significant” room sharing is considered (i.e. when at least one room occupant is during an infectious period, defined as the period between 3 days before onset of symptoms and 14 days after). Of the 319 days where a room was shared, 256 days (80.3%) involved a room being shared by 2 patients, and 63 (19.7%) by 3 patients. The proportion of patient-to-patient transmissions that involved patients sharing a room was calculated for each posterior transmission tree (by determining whether the ancestor/infector had ever shared a room with the infectee), with the denominator being all patient-to-patient transmissions.

When responding to this comment we noticed an error in the pre-processing stage of contact data (n = 839 contacts, 4%, that should not have been included). We re-ran all the analyses after correcting this error. This has led to minor changes in our numeric results does not alter our major findings and conclusions. We apologise for this inconvenience, and thank the referee for their in-depth review which has given us the opportunity to double-check our work.

– There is a long-tailed distribution of infection by each individual (Figure 4). How much do heavy spreaders influence the results e.g. would the results from Figure 5 all hold if individual superspreader events were removed from the dataset?

We thank the Reviewer for this interesting observation and question. Altogether, the proportion of super-spreading events (defined as generating ≥5 secondary cases) among all posterior trees is very low (0.4%). The majority of posterior trees (54.1%) have no infectors with ≥5 secondary cases. The proportion of posterior trees with 1, 2, and ≥3 infectors with ≥5 secondary cases (super-spreaders) are 37.3%, 7.9% and 0.7%, respectively. The overwhelming majority of posterior trees (88.6%) do not have infectors who are the plausible ancestors of ≥6 infectees. For these reasons, we anticipated the influence of the superspreading events across all posterior transmission trees to be negligible. To confirm this, we performed a sensitivity analysis, estimating the proportion of transmissions (*f_case_*) attributed to each case type (HCW_covid_, HCW_outbreak_, patient_noso_), i.e., the results from Figure 5, but excluding posterior transmission trees where there is at least one infector with ≥5 secondary cases. In Author response image 1, we show the original figure 5 and in Author response image 2, the corresponding results from our sensitivity analysis. Results are very similar. Should the editor find it useful, we would be happy to add this sensitivity analysis to the Appendix.

**Author response image 1. sa2fig1:** 

– When calculating expectation in Figure 5, this is calculated by the prevalence of Covid-19 in each type of individual on a given ward. In our understanding, the ward was not considered here. Could the authors justify that choice? Would it be feasible to calculate an expectation which took into account the higher risk of transmission when sharing a room, and use this as the comparator against observed infections?

The scenario chosen for the comparison is the most agnostic in that it does not make any assumptions regarding contact patterns. Instead, it assumes that all individuals are able to infect each other with the same probability, and, as a consequence, a group is able to infect an individual based on the proportion of individuals belonging to that group. However, this may be seen as an upper bound of the effect, as indeed it does not account for ward/room sharing in addition to the case type. Indeed, the analysis estimates excess transmission, but it does not specify what the excess is due to. For this reason, we performed the analysis of the ward type as well as the ward-to-ward matrix. Unfortunately, it is not feasible to account for both ward/room sharing and case type, as the combination of multiple characteristics would decrease the sample size and may potentially lead to type II error. As highlighted by the Reviewer, to calculate an expectation which takes into account the higher risk of transmission by sharing a room, we would need to estimate this, yet there is little data to inform this parameter.

We apologise but there was a small error in the Legend for Figure 5: instead of “*f_ward_*”, the legend should read “*f_case_*”. We have now corrected this.

– Page 11 line 231, the referenced figure doesn't indicate these probabilities. What is the source?

We thank the Reviewer for pointing this out. Indeed, the Figure was intended to show the ward movements of the patients and the dates where the patients overlapped. The probabilities of infection were derived from the output of the model (basically from Figure 3). We have added that the referenced figure shows the ward movements, hoping this makes it clearer.

– How was the date of April 9 fixed to define a cut-off between the early phase and late phase? Is it just based on the peak incidence? Or due to a change in screening practice?

The epidemic phase was determined on the basis of several factors. First, many preventive measures were implemented around that date (e.g., weekly screening of patients on April 07, enhanced testing of HCWs on April 09, decrease in bed-occupancy and closures on April 11). Second, approximately half of the cases occurred in each phase of the outbreak. Third, the overall trend in the epidemic curve can be interpreted as increasing in the first phase, and decreasing in the second phase.

– The large outbreak size makes figure 3 quite hard to interpret for the reader. Would it be possible to improve it possibly by zooming on some specific cases or wards, and adding global statistics informing on the precision/quality of the inference of transmission routes?

We agree with Reviewer 1 that Figure 3 was difficult to read. We have now created an interactive version of this figure where one can hover over points and zoom in selected areas. We have also removed the colours, which did not have a particular meaning. We have decided to move this new interactive figure as Figure 3 – Supplementary Figure 1, and have replaced it with a more informative figure demonstrating the extent to which our model is able to infer of transmission routes (i.e., showing the distribution of the posterior support of ancestries based on different criteria (individual, case type, ward), see new Figure 3).

– How does the number of secondary infections relate to the population's sizes (i.e. at-risk patients and HCWs from the studied wards). Is that relationship expected?

The Reviewer raises a valid and interesting point. Generally, and under normal circumstances, the ward sizes are comparable, as are the number of HCWs working in each ward. However, the number of patients in a ward is constantly fluctuating due to admissions/discharges as well as ward closures. Similarly, the number of HCWs fluctuates due to sick leave or annual leave. Therefore, the wards are not “closed systems”, and as the denominator fluctuates, more complex models would be required in order to explain the underlying mechanisms. We have added a sentence in the discussion to acknowledge this limitation:

“We were unable to relate the number of secondary infections with the population (ward) size; more complex models would be required to explain the underlying mechanisms in a context of fluctuating denominators, e.g., due to ward closures, etc.”

– We may have missed this: (apologies if so): were HCWs cohorted by wards or were there HCWs shared among wards? How may that affect the results here?

We thank the Reviewer for pointing this out. HCWs did work exclusively in one ward type or another. We have clarified this in the Methods (“Definitions” section):

“HCWs worked in either one or another type of ward, except for one HCW who worked in both Covid and non-Covid wards (although the proportion and/or days worked in each type of ward is unknown). Six HCWs worked across multiple wards (e.g. on-call) and were attributed to Covid-wards for the analyses.”

Reviewer #2 (Recommendations for the authors):This paper addresses an important question about the dynamics of SARS-CoV-2 in a geriatric acute-care hospital. This reviewer's comments address primarily manuscript clarity.General Comments:Patients should not be equated with their conditions. Please replace Covid-19 and non-Covid 19 patients with patients who have Covid-19 and patients who do not have Covid-19 throughout the manuscript.Compare takes "with" not "to".

We absolutely agree with the reviewer that patients should not be equated with their conditions. Nevertheless, we feel that the use of “patients who have Covid-19” may unnecessarily lengthen the manuscript and complicate reading. Also, we believe that the use of “Covid-19 patients” is neither disrespectful or stigmatising, and has been used in official documents from highly respected institutions (e.g., WHO, https://apps.who.int/iris/rest/bitstreams/1292529/retrieve; NHS England, https://www.england.nhs.uk/coronavirus/publication/assessment-monitoring-and-management-of-symptomatic-covid-19-patients-in-the-community/). Nevertheless, should the Editor feel the same way as the Reviewer, we would be happy to reword.

We have corrected “compared to” with “compared with”.

The manuscript includes many "there is," "there are," "it is," xxx. These statements are in passive voice and are usually wordier and more convoluted than a similar statement in active voice. This reviewer suggests revising most or all of these statements. For example, the following sentence could be revised as shown."The reservoir of SARS-CoV-2 in healthcare environments may contribute to amplifying the pandemic [5], and as such, it is important to understand transmission dynamics in these settings.""Because the reservoir of SARS-CoV-2 in healthcare environments may contribute to amplifying the pandemic [5], we need to better understand transmission dynamics in these settings."

We thank the Reviewer for their comment, which we have taken into consideration wherever possible. We do, however, defer to the Editor and the Editorial team for further editorial and wording matters.

Statements like "in order to" and "as well as" have extra words that don't add to the meaning. They can be shortened to "to" and "and".

We understand the Reviewer’s point, and have modified accordingly.

Specific CommentsLine 67: Is the word "unexpectedly" needed? This reviewer was confused by that statement. I would assume HCWs knowing that they were caring for patients with Covid-19 might be more in adherence with PPE use and hand hygiene than those on non-Covid-19 units.

We understand the Reviewer’s point, and indeed, in 2022 it is conventional wisdom to make the assumption raised. However, at the time these outbreaks occurred, we did not think there would be such a stark difference in terms of cross-infections amongst HCWs in Covid vs. non-Covid wards. Indeed, the finding is unexpected. For this reason, should the Editor agree, we prefer to keep the word “unexpectedly”.

Lines 93-95: The meaning of "in nosocomial Covid-19" is vague and should be clarified. Does this mean HCWs' role in transmitting SARS-CoV-2 within healthcare facilities is complex?Does it mean HCWs role in the epidemiology of nosocomial SARS-CoV-2 infection iscomplex? This reviewer does not like the word "victim" (although the alteration is nice) in this setting. This reviewer thinks the last part of this sentence will work better if revised to: "as HCWs can acquire SARS-CoV-2 in the community or from their peers and patients and they can transmit SARS-CoV-2 to peers and patients [7, 8].

We thank the Reviewer for their comment. We have modified the sentence “in nosocomial Covid-19” to “in nosocomial Covid-19 transmission dynamics”, and hope that the sentence is clearer. We understand that the reviewer does not like the word victim; however, we believe that its use is justified in the present context, and unlike many health conditions, we feel that in the case of Covid-19 in most societies around the world there is no particular stigma associated with it. We also respectfully refer the Reviewer to the book entitled “The Patient as Victim and Vector: Ethics and Infectious Disease” by Margaret P. Battin, Leslie P. Francis, Jay A. Jacobson, and Charles B. Smith (Oxford Scholarship Online, 2009. DOI:10.1093/acprof:oso/9780195335842.001.0001).

MethodsThe authors switch frequently between active and passive voice. This reviewer suggests picking one and sticking with it. This reviewer prefers an active voice for clarity and brevity.

We thank the Reviewer for these suggestions that we have implemented.

Line 162: This reviewer does not understand what the authors mean by "trajectories" in this sentence.

We meant “ward movements”, which has now replaced “trajectories”.

Lines 165-6: Revise to something like: "conservative in that it accounts for non-infectious contacts and incomplete reporting of infectious contacts and it estimates the proportion of the total contacts in these categories."

We have replaced the sentence to the following, hoping that the Reviewer finds it clearer:

“The manner by which outbreaker2 handles these contacts is conservative in that it allows for non-infectious contacts to occur (false positives) and incomplete reporting of infectious contacts (false negatives). In addition, the model estimates the proportions of the proportions of these contacts.”

Lines 18-91: Suggested revision. A post hoc epidemiological analysis of the 53 patients found that 4 of these patients likely acquired Covid-19 in the community (CA-Covid-19). The 49 nosocomial cases represented 20.2% (49/242) of all patients with COVID-19, and 81.7% (49/60) of nosocomial Covid-19 cases identified in the Geriatric Hospital. The ward-level attack rates ranged from 10 to 19% among patients and 21% of all HCWs in the geriatric..."

We thank the Reviewer for these suggestions that we have implemented.

Line 203: This reviewer does not understand this sentence given that the verb appears to be "shows".

We have modified this sentence in order for it to be grammatically correct, and hopefully easier to understand:

“There is also a large cluster (BS 68% with signature mutations C8293T, T18488C, and T24739C) with several subclusters includes 19 HCWs, nine patients, and three community cases; ward movements for the patients are shown in Appendix Figure 1”

Lines: 273-5: Suggest revising to something like: "Fourth, we identified multiple importation events that led to a substantial number of secondary cases or clusters. Most transmission events were related to HCWs, but one (?? or some) were related to a patient with community-acquired Covid-19." Note: the authors did not make clear how many transmissions were related to this patient.

We thank the Reviewer for these suggestions that we have implemented. We would like to point out that the number of secondary transmissions related to the patient with community-acquired Covid-19 was mentioned in the Results, under the heading “Imported cases”, which, incidentally, has been amended to better reflect the uncertainty associated with the reconstruction of the transmission chains:

“Imported Cases

From the reconstructed trees, we identified 22 imports in total (17 HCWs, five patients) with posterior support ≥10%. The 22 imported cases generated 41 secondary cases (posterior support ≥10%), with a median posterior support of 32.4% (IQR 17.0-53.7%). When restricting to imports with ≥50% posterior support, there were 16 imported cases 16 (12 HCWs, four patients), generating 35 secondary cases. There was some degree of uncertainty, reflected by circular transmission pathways, in determination of imported cases and their secondary cases. There were 6 transmission pairs (C114-C115, C153-H1057, H1008-H1059, H1011-H1019, H1017-H12021, H1052-H1082) where there was uncertainty as to which of the cases was imported and which was a secondary case, i.e. for each case in the pair there was a ≥10% probability of importation, but also ≥10% probability of being a secondary case of an imported case. Therefore, in total, 29 cases were “pure” secondary cases of imported cases (Table 3).”

Lines 282-292: Suggest revising to something like: "HCWs' behavior may have affected transmission. Indeed, Ottolenghi et al., found that HCWs caring for Covid-19 patients were concerned about becoming infected while caring for patients, and therefore may apply IPC measures more rigorously than when caring for patients who do not have Covid-19 [23]. HCWs working in non-Covid wards may not have felt threatened by Covid-19 patients on other wards. HCWs in non-Covid wards also may have underestimated the transmission risk from their peers to a greater extent than those working in Covid wards, and thus did not maintain physical distancing well. In addition, we found a higher mean duration (2.9 days) of presenteeism despite symptoms compatible with Covid-19 among HCWs working in non-Covid wards than for those working in Covid wards (1.6 days), which gives credence to the abovementioned possible explanations for the different transmission patters. Other factors (e.g., work culture, baseline IPC practices) also may have affected transmission patterns.This reviewer wonders if the following sentence is necessary? "HCWs working in non-Covid wards may not have felt threatened by Covid-19 patients on other wards."

We thank the Reviewer for taking the time to make these detailed suggestions. We have largely changed the paragraph accordingly:

“HCWs' behaviour may have affected transmission. Indeed, Ottolenghi et al., found that HCWs caring for Covid-19 patients were concerned about becoming infected while caring for patients, and therefore may apply IPC measures more rigorously than when caring for patients who do not have Covid-19 [23]. HCWs working in non-Covid wards may not have felt threatened by Covid-19 patients who were in principle allocated to other wards, and thus not in their direct care. HCWs in non-Covid wards also may have underestimated the transmission risk from their peers to a greater extent than those working in Covid wards, and thus may not have maintained physical distancing well. In addition, we found a higher mean duration (2.9 days) of presenteeism despite symptoms compatible with Covid-19 among HCWs working in non-Covid wards than for those working in Covid wards (1.6 days), which gives credence to the abovementioned possible explanations for the different transmission patterns. Other factors (e.g., work culture, baseline IPC practices) also may have affected transmission patterns.”

Line 294: This author is wondering how the same ward is close proximity when patients are not in the same room. How would the patients have contact with each other?

The architectural layout of the hospital means that there are 2 wings per floor, each with a central corridor, and 2 wards per wing (without separation). On either side of the central corridor are patient rooms and/or offices (nursing, medical, etc.). Therefore, the wards themselves are crowded areas, and it is sufficient for a patient to step outside their room in the corridor to potentially come in contact with another patient. We have added this information in the Appendix, alongside data on room sizes, under the heading “Ward architecture and room sizes in the geriatric hospital”.

Lines 294-309: Suggested revisions: We cannot exclude transmission from a “point-source” or via an HCW’s contaminated hands (i.e., an unidentified HCW who transmitted SARS-CoV-2 to multiple patients in the same ward). To date, we have little evidence to suggest that this is the case; indeed, the transmission patterns were robust to changes in the model assumptions. Mathematical models have suggested that single-room isolation of suspected cases could potentially reduce the incidence of nosocomial SARS-CoV-2 transmission by up to 35% [24]. In the outbreaks we describe, symptomatic patients were identified promptly, with a median delay of 0 days between symptom onset and their first positive test. However, these precautions may not be sufficient as patients may transmit the virus when they are pre-symptomatic [25]. Thus, infection prevention teams may need to identify patients at high risk of developing nosocomial Covid-19 [26] if single rooms are not available for all exposed patients (e.g., in cases of overcrowding). For example, Mo et al., found that exposure to community-acquired cases who were identified and segregated or cohorted was associated with half the risk of infection compared with exposure to hospital-acquired cases or HCWs who may be asymptomatic [27]. One possible explanation for this finding is that patients with CA-Covid may have passed the peak of infectiousness when they are admitted, whereas patients with HA-Covid cases have frequent unprotected contact with HCWs and other patients during their period of peak infectiousness.

We thank the Reviewer for taking the time to make these detailed suggestions, which we have implemented in the manuscript.

Line 310: Suggested edit: The current evidence does not support the use of real-time genomics for control of SARS-CoV-2 nosocomial outbreaks [28].

We thank the Reviewer for their suggestion. We have also taken the opportunity to include a reference to the preprint by Stirrup et al., (medRxiv 2022.02.10.22270799) which reports the COG-UK HOCI study.

Lines 324-328: Suggested edits: Despite these strengths, our study had some limitations. First, we included only one sequence from a CA-Covid. However, the method we used to reconstruct who infected who is able to cope with and identify missing intermediate cases, which allowed us to estimate that our sample included the overwhelming majority of cases (91.5%). In addition, we performed these investigations during the first pandemic wave in a susceptible population...."

We thank the Reviewer for these suggestions, some of which we have implemented.

Line 334: Replace "in the case of" with "during".

We believe that “in the case of” is a better fit, as although it is likely that there will be another pandemic, it is still an unknown.

Lines 336-338: The main point of the conclusion is not clear. What does it mean to take into account the complex interplay between HCWs in dedicated Covid-19 wards vs non-Covid wards? The way this is worded makes it sound like the HCW on the two different kinds of wards are interacting, which likely is not what the authors intended.

We have replaced the conclusion by the following sentence, hoping it is clearer:

“In conclusion, strategies to prevent nosocomial SARS-CoV-2 transmission in geriatric settings should take into account the potential for patient-to-patient transmission and the transmission dynamics between HCWs in non-Covid wards, which our study suggests may differ from those in dedicated Covid-19 wards.”

Table 1: This reviewer doesn't understand what is meant by "Onset of symptoms before swab date, n (%)". Is this the number of people who were symptomatic before they were tested?

Yes, the Reviewer is correct.

Table 2: Should HCWs be written out in the title? Readers in the US will likely not know what a logopedist is. We will call this person a speech therapist. Perhaps the authors could include a footnote defining this term. Did the authors address the longer time from onset of symptoms to testing for HCWs on non-Covid wards compared with those on Covid wards in the results and discussion?

HCWs was written out in full. Logopedist was replaced by “speech therapist”.

The longer time from onset of symptoms to testing has been added in the Results section. We had already brought up the issue in the third paragraph of the Discussion.

Table 3: Dose "Secondary onward transmission" refer to transmission from the case in the first column? Consider putting NA (not applicable) in the blank cells.

We have replaced “Secondary onward transmission” by “Secondary onward transmission by imported case” in order to make it clearer. Blank cells were replaced by N/A.

Line 515: Suggested edit: "Histograms displaying the distributions of secondary cases...."

Thank you for this suggestion which we have taken into account.

Line 525: Is something missing between "... patient community and "proportion"?

Apologies as the phrase “proportion of infections attributed to the type of case” was accidentally inserted. We hope that after deletion the sentence is clearer.

Reviewer #3 (Recommendations for the authors):Methods:The authors present the IPC measures on page 6 in the main text and in the supplementary material. However, there seems to be no information on how HCWs adhered to the recommendations that were not mandatory and whether adherence differed between wards. This is probably because the authors did not have any information on that. This is a limitation and should be explained more explicitly or highlighted in the discussion.

We agree with the Reviewer that this is a limitation of our study. We have added this in the Discussion:

“We did not collect data on adherence to Covid-specific IPC recommendations by HCWs in different wards.”.

It's not clear to me whether the HCWs only worked in either COVID or non-COVID wards during the study period. Please clarify in the Methods section.

We thank the Reviewer for pointing this out. HCWs did work exclusively in one ward type or another. We have clarified this in the Methods (“Definitions” section):

“HCWs worked in either one or another type of ward, except for one HCW who worked in both Covid and non-Covid wards (although the proportion and/or days worked in each type of ward is unknown). Six HCWs worked across multiple wards (e.g. on-call) and were attributed to Covid-wards for the analyses.”

On page 7, line 139: The authors state that they substituted the missing date of symptom onsets with the median difference between symptom onset date and swab date? The epidemic curve in Figure 1 shows the incidence on the y-axis but does the x-axis (date of symptom onset) make sense if not all of the cases were symptomatic?

The Reviewer has rightly pointed out that the Epidemic curve is based on symptom onset and that for the asymptomatic cases (n = 8, 4.4%) there is no date available. It is common practice to use date of symptom onset to construct epidemic curves, and this explains our choice. So yes, the epidemic curve includes the asymptomatic cases, and we have clarified this in the legend of Figure 1 (“Includes 8 asymptomatic cases for whom date of onset was inferred (c.f. text).”). We could have, instead, used the date of the first positive sample. While using the date of the first positive sample might ensure that there were no missing data points, it does provide a completely different picture of the progression of the outbreak. For example, the earliest positive swab was on March 11, 2020, whereas the earliest onset of symptoms was 6 days earlier on March 05, 2020. An additional reason is that the outbreaker2 model uses date of symptom onset as the starting point from which the date of infection is inferred statistically. Should the Editor request that we construct an epidemic curve using date of swab or removing the patients with the missing date of symptom onset, we will gladly do it. We can also highlight in the epidemic curve the 8 asymptomatic cases, should the Editor and/or the Reviewer request this.

Page 8, line 169 and Figure 4: How was the epidemic phase determined?

The epidemic phase was determined on the basis of several factors. First, many preventive measures were implemented around that date (e.g., weekly screening of patients on April 07, enhanced testing of HCWs on April 09, decrease in bed-occupancy and closures on April 11). Second, approximately half of the cases occurred in each phase of the outbreak. Third, the overall trend in the epidemic curve can be interpreted as increasing in the first phase, and decreasing in the second phase. We have added justification for this in the Appendix.

Results:At the beginning of the Results section, the authors write that post hoc epidemiological analysis showed that were likely 4 patients that had a community-acquired infection. What exactly did the post-hoc epidemiological analysis entail? Later on (page 10, lines 206ff), they present their analysis of the imported cases. How are these related?

We thank the Reviewer for their careful reading of our paper. The post hoc epidemiological analysis consisted of a chart review performed by the investigators at the start of the study, as opposed to conclusions that were arrived at during the management of the outbreak. The classification of cases as being imported is based on the model output, and they are defined as those that do not have apparent ancestors among the cases included in the outbreak. We have included this information in the manuscript, hoping that this has been clarified.

Discussion:Could the authors discuss whether misclassifications (either of community- or hospital-acquired cases) even after the analysis is still possible and what impact it could have on their results?

The Reviewer rightly points out that misclassifications may still occur. It is even part of the method (outbreaker2) to quantify the uncertainty in determining the ancestry, as it operates within a Bayesian framework. Indeed, in most cases (c.f. new Figure 3) there are very few patients for whom the ancestry is identified with 100% posterior probability. For example, the model predicted that patients C107 and C115 were most likely imported cases with posterior probabilities of 100% and 57.5%, respectively. The predicted dates of infection were March 14-22 for case C107 and March 8-24 for case C115, which was after their admission dates (March 3 and March 18, respectively). The probability that infection occurred on or after date of admission for case C115 was 81%. Therefore the model acknowledges the uncertainty in the classification of these cases. We have modified the section on imported cases (under the heading “imported cases”), and added a sentence after referencing the new Figure 3 to highlight the uncertainty in the model.

Finally, the fact that a case has a probability of being “imported” does not preclude the fact that it is still nosocomial cases, simply that their infector was not identified in this outbreak c.f. update in the manuscript (lines 241-259).

SupplementPage 5: Could the authors elaborate on "global influence" for imported cases?

The global influence is a way of measuring the impact of a single observation on the log-likelihood of a parameter. During a preliminary run of the model, cases are removed in turn (with a leave one out approach) and the “global influence” of each case is measures as the extent to which removing it affects the genetic log-likelihood. Cases with high global influence (whose removal dramatically affects the genetic likelihood) are considered to be genetic outliers, and classified as imported cases. This approach is described in more detail in the original outbreaker manuscript (Jombart et al., PLoS Comput Biol 10(1): e1003457). We have added these details in the Appendix.